# Towards a Theoretical Understanding of Memorization in Diffusion Models

## Abstract

As diffusion probabilistic models (DPMs) are being employed as mainstream models for Generative Artificial Intelligence (GenAI), the study of their memorization of training data has attracted growing attention. Existing works in this direction aim to establish an understanding of whether or to what extent DPMs learn via memorization. Such an understanding is crucial for identifying potential risks of data leakage and copyright infringement in diffusion models and, more importantly, for trustworthy application of GenAI. Existing works revealed that conditional DPMs are more prone to training data memorization than unconditional DPMs, and the motivated data extraction methods are mostly for conditional DPMs. However, these understandings are primarily empirical, and extracting training data from unconditional models has been found to be extremely challenging. In this work, we provide a theoretical understanding of memorization in both conditional and unconditional DPMs under the assumption of model convergence. Our theoretical analysis indicates that extracting data from unconditional models can also be effective by constructing a proper surrogate condition. Based on this result, we propose a novel data extraction method named **Surrogate condItional Data Extraction (SIDE)** that leverages a time-dependent classifier trained on the generated data as a surrogate condition to extract training data from unconditional DPMs. Empirical results demonstrate that our SIDE can extract training data in challenging scenarios where previous methods fail, and it is, on average, over 50% more effective across different scales of the CelebA dataset.

## 1 Introduction

The diffusion probabilistic models (DPMs) (Ho et al., 2020; Sohl-Dickstein et al., 2015; Song & Ermon, 2019) is a family of powerful generative models that learn the distribution of a dataset by first gradually destroying the structure of the data through an iterative forward diffusion process and then restoring the data structure via a reverse diffusion process. Due to their outstanding capability in capturing data distribution, DPMs have become the foundation models for many pioneering Generative Artificial Intelligence (GenAI) products such as Stable Diffusion (Rombach et al., 2022), DALL-E 3 (Betker et al.), and Sora (Brooks et al., 2024). Despite the widespread adoption of DPMs, a potential risk they face is *data memorization*, i.e., the risk of memorizing a certain proportion of the raw training samples. This could result in the generation of memorized (rather than new) samples via direct copying, causing data leakage, privacy breaches, or copyright infringement, as highlighted in the literature (Somepalli et al., 2022; 2023; Asay, 2020; Cooper & Grimmelmann, 2024). A current case argues that Stable Diffusion is a 21st-century collage tool, remixing the copyrighted creations of countless artists whose works were included in the training data(Butterick, 2023). Furthermore, data memorization also gives rise to data extraction attacks, which is one type of privacy attacks that attempt to extract the training data from a well-trained model. Notably, recent work by Carlini et al. (2023) demonstrated the feasibility of extracting training data samples from DPMs like Stable Diffusion (Rombach et al., 2022), revealing the potential dangers associated with these models.

Several works have investigated the data memorization phenomenon in diffusion models. For example, it has been observed that there exists a strong correlation between training data memorization and conditional DPMs, i.e., being conditional is more prone to memorization (Somepalli et al., 2023). Gu et al. (2023) investigates the influential factors on memorization behaviors via a comprehensive set of experiments. They found that conditioning on random-labeled data can significantly trigger

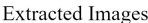 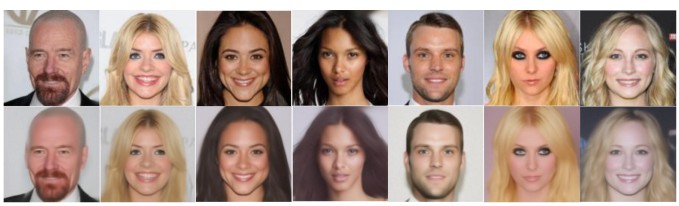

Figure 1: A few examples of the extracted images from a DDPM trained on a subset of the CelebA dataset using our SIDE method. *Top*: training images; *bottom*: extracted images.

memorization, and unconditional models memorize much less training data. Though inspiring, these understandings are primarily empirical. Moreover, without a unified theoretical understanding of memorization in both conditional and unconditional DPMs, it is extremely difficult to design an effective data extraction method for unconditional DPMs.

In this work, we first introduce a memorization metric to quantify the degree of memorization in DPMs by measuring the overlap between the generated and training data in a point-wise manner. Based on this metric, we present a theoretical framework that explains why conditional generative models memorize more data and why random labeling can lead to increased memorization. Our theoretical analysis indicates that a classifier trained on the same or similar training data can serve as a surrogate condition for unconditional DPMs. By further making the classifier time-dependent (to the diffusion sampling process), we propose a novel data extraction method named **Surrogate condItional Data Extraction (SIDE)** to extract training data from unconditional DPMs. We empirically verify the effectiveness of SIDE on CIFAR-10 and different scales of the CelebA dataset (attack results in Figure 1), confirming the accuracy of the theoretical framework.

In summary, our main contributions are:

- We introduce a novel metric to measure the degree of point-wise memorization in DPMs and present a theoretical framework that explains 1) why conditional DPMs are more prone to memorization, 2) why random labels can lead to more memorization, and 3) implicit labels (e.g., the learned clusters) can serve as a surrogate condition for unconditional DPMs.

- Based on our theoretical understanding, we propose a novel training data extraction method **SIDE** that leverages the implicit labels learned by a time-dependent classifier to extract training data from unconditional DPMs.

- We evaluate the effectiveness of SIDE on CIFAR-10 and various scales of CelebA datasets, and show that, on average, it can outperform the baseline method proposed by Carlini et al. (2023) by more than 50%.

## 2 RELATED WORK

**Diffusion Probabilistic Models**  DPMs (Sohl-Dickstein et al., 2015) such as Stable Diffusion (Rombach et al., 2022), DALL-E 3 (Betker et al.), Sora (Brooks et al., 2024), Runway (Rombach et al., 2022), and Imagen (Saharia et al., 2022) have achieved state-of-the-art performance in image/video generation across a wide range of benchmarks (Dhariwal & Nichol, 2021). These models can be viewed from two perspectives. The first is score matching (Song & Ermon, 2019), where diffusion models learn the gradient of the image distribution (Song et al., 2020). The second perspective involves denoising DPMs (Ho et al., 2020), which add Gaussian noise at various time steps to clean images and train models to denoise them. To conditionally sample from diffusion models, (Dhariwal & Nichol, 2021) utilizes a classifier to guide the denoising process at each sampling step. Additionally, (Ho & Salimans, 2022) introduces classifier-free guidance for conditional data sampling using DPMs.

**Memorization in Diffusion Models**  Early exploration of memorization in large models was focused on language models (Carlini et al., 2022; Jagielski et al., 2022), which has motivated more in-depth research on image-generation DPMs (Somepalli et al., 2023; Gu et al., 2023). Notably, a recent research on image-generation DPMs. Somepalli et al. (2022) found that 0.5-2% of generated images duplicate training samples, a result concurrently reported by Carlini et al. (2023) in broader

experiments for both conditional and unconditional diffusion models. Further studies Somepalli et al. (2023) and Gu et al. (2023) linked memorization to model conditioning, showing that conditional models are more prone to memorization. To address this issue, anti-memorization guidance has been proposed to mitigate memorization during the sampling process Chen et al. (2024a). So far, the motivated data extraction attack or defend methods from these empirical understandings are mostly for conditional DPMs Carlini et al. (2023); Webster (2023), and studies have shown that extracting training data from unconditional DPMs can be much more challenging than that on conditional DPMs Gu et al. (2023). Although the empirical understandings are inspiring, a theoretical understanding of the memorization behaviors of DPMs is missing from the current literature. A recent attempt by Ross et al. (2024) uses the local intrinsic dimension (LID) metric to characterize memorization in DPMs. However, the finding that lower LID leads to more memorization is only verified on a toy 1-dimensional dataset and only a few generated images, and it fails to address the data extraction challenge on unconditional DPMs. This challenge is exemplified by the brute-force methods currently employed, as discussed in (Somepalli et al., 2022; Carlini et al., 2023), highlighting the inadequacies of existing approaches for extracting data from unconditional diffusion models. In response to these challenges, we introduce a theoretical framework to characterize memorization in DPMs, which further motivates a novel data extraction method for unconditional DPMs.

## 3 PROPOSED THEORY

In this section, we introduce a memorization metric and provide a theoretical explanation for the universality of data memorization in both conditional and unconditional diffusion models.

### 3.1 MEMORIZATION METRIC

Intuitively, the memorization of fixed training data points (i.e., pointwise memorization) can be quantified by the degree of overlap between the generated distribution and the distributions centered at each training data point. Given a generative model and training dataset, we propose the following memorization metric to quantify the memorization of the training data points in the model.

**Definition 1 (Pointwise Memorization)** *Given a generative model $p_\theta$ with parameters $\theta$ and training dataset $\mathcal{D} = \{\boldsymbol{x}_i\}_{i=1}^N$, the degree of memorization in $p_\theta$ of $\mathcal{D}$ is defined as:*

$$\mathcal{M}_{point}(\mathcal{D}; \theta) = \sum_{\boldsymbol{x}_i \in \mathcal{D}} \int p_\theta(\boldsymbol{x}) \log \frac{p_\theta(\boldsymbol{x})}{q(\boldsymbol{x}, \boldsymbol{x}_i, \epsilon)} \, d\boldsymbol{x}, \tag{1}$$

*where $\boldsymbol{x}_i \in \mathbb{R}^d$ is the $i$-th training sample, $N$ is the total number of training samples, $p_\theta(\boldsymbol{x})$ represents the probability density function (PDF) of the generated samples, and $q(\boldsymbol{x}, \boldsymbol{x}_i, \epsilon)$ is the probability distribution centered at training sample $\boldsymbol{x}_i$ within a small radius $\epsilon$.*

A straightforward choice for $q(\boldsymbol{x}, \boldsymbol{x}_i, \epsilon)$ is the Dirac delta function centered at training data point $\boldsymbol{x}_i$: $q(\boldsymbol{x}, \boldsymbol{x}_i, \epsilon) = \delta(\boldsymbol{x} - \boldsymbol{x}_i)$. However, this would make Eq. (1) uncomputable as the Dirac delta function is zero beyond the $\epsilon$-neighborhood. Alternatively, we could use the Gaussian distribution with a covariance matrix $\epsilon I$ ($I$ is the identity matrix) for $q(\boldsymbol{x}, \boldsymbol{x}_i, \epsilon)$:

$$q(\boldsymbol{x}, \boldsymbol{x}_i, \epsilon) = \frac{1}{\sqrt{(2\pi\epsilon)^d}} \exp\left\{-\frac{1}{2\epsilon}(\boldsymbol{x} - \boldsymbol{x}_i)^\top(\boldsymbol{x} - \boldsymbol{x}_i)\right\}. \tag{2}$$

Note that in Eq. (1), a smaller value of $\mathcal{M}_{point}(\mathcal{D}; \theta)$ indicates more overlap between the two distributions and thus *more memorization*. As $\epsilon$ tends to 0, the measured memorization becomes more accurate, with the limit characterizing the intrinsic memorization capability of model $p_\theta$.

**Semantic Memorization** As a metric, $\mathcal{M}_{point}(\mathcal{D}; \theta)$ should be monotonous with the actual semantic memorization effect, which refers to the model's tendency to reproduce unique features of the training samples rather than generating near-duplicate examples. Here, we define semantic memorization in the latent space as it distills the unique semantic information of each training sample.

**Definition 2 (Semantic Memorization)** *Let $\mathcal{D} = \{\boldsymbol{x}_i\}_{i=1}^N$ be the dataset, $p_\theta(x)$ be the PDF of the generated samples by model $p_\theta$, and $\boldsymbol{z}$ be the learned latent code for data sample $\boldsymbol{x}$. The semantic*

*memorization of model $p_\theta$ on $\mathcal{D}$ is:*

$$\mathcal{M}_{semantic}(\mathcal{D};\theta) = \sum_{\boldsymbol{z_i} \in \boldsymbol{D}} \int p_\theta(\boldsymbol{z})(\boldsymbol{z}-\boldsymbol{z_i})^T(\boldsymbol{z}-\boldsymbol{z_i})\,d\boldsymbol{z}, \qquad (3)$$

*where $\boldsymbol{z_i}$ denotes the ground truth latent code of $\boldsymbol{x_i}$.*

$\mathcal{M}_{semantic}(\mathcal{D};\theta)$ measures semantic memorization because it evaluates how well the learned representations $\boldsymbol{z}$ of training samples align with their ground truth latent codes, capturing the underlying structure of the data rather than simply memorizing the exact duplicates of the training data points. In practice, the ground truth latent codes are known but can be approximated by an independent encoder.

The following theorem formulates the relationship between pointwise and semantic memorization.

**Theorem 1** *Pointwise memorization $\mathcal{M}_{point}(\mathcal{D};\theta)$ is monotonic to semantic memorization, formally:*

$$\frac{\partial \mathcal{M}_{point}(\mathcal{D};\theta)}{\partial \mathcal{M}_{semantic}(\mathcal{D};\theta)} = \frac{1}{2\epsilon} + \frac{\mathrm{Tr}(\Sigma_{p_\theta}^{-1})}{2} > 0 \qquad (4)$$

*where $\Sigma_{p_\theta}$ is the covariance matrix of the learned latent distribution of the training data.*

We first build the relationship map $\mathcal{M}_{point}$ into latent space, and then break down Eq. (4) into separate components and derive each component separately. So, the Eq. (4) can be simplified to $\frac{1}{2\epsilon}$. The detailed proof can be found in Appendix A.2. Note that the memorization effect studied in this work refers specifically to the pointwise memorization $\mathcal{M}_{point}$.

### 3.2 THEORETICAL FRAMEWORK

Based on pointwise memorization, here we present a theoretically framework that explains why conditional DPMs memorize more data. Our theoretical framework is based on the concept of *informative labels*, which refers to information that can differentiate subsets of data samples. We first give a formal definition of informative labels and then prove their two key properties: 1) they facilitate tighter clustering of samples around their respective means, and 2) they reduce variance in the latent representations. Building upon the two properties, we theoretically show that conditional DPMs memorize more data.

**Informative Labels** The concept of *informative labels* has been previously discussed as class labels (Gu et al., 2023). In this work, we introduce a more general definition that encompasses both class labels and random labels as special cases. We define an informative label as follows.

**Definition 3 (Informative Label)** *Let $\mathcal{Y} = \{y_i, y_2, \cdots, y_C\}$ be the label set for training dataset $\mathcal{D}$ with $C$ unique labels. $y_i$ is the associated label with $\boldsymbol{x_i}$, and $\mathcal{D}_{y=c} = \{\boldsymbol{x_i} : \boldsymbol{x_i} \in \mathcal{D}, y_i = c\}$ is the subset of training samples shared the same label $y = c$. A label $y = c$ is an informative label if*

$$|\mathcal{D}_{y=c}| < |\mathcal{D}|. \qquad (5)$$

Here, the labels are not limited to the conventional class labels; they can also be text captions, features, or cluster information that can be used to group the training samples into subsets. The above definition states that an information label should be able to differentiate a subset of samples from others. An extreme case is that all samples have the same label; in this case, the label is not informative. According to our definition, class-wise and random labels are special cases of informative labels. Informative labels can be either *explicit* like class/random labels and text captions, or *implicit* like silent features or clusters. Next, we will explore the correlation between informative labels and the clustering effect in the latent space of a generative model with an encoder and decoder. In diffusion models, the encoder represents the forward diffusion process, while the decoder represents the reverse process. Notably, the latent space can be defined at earlier timesteps according to (Ho et al., 2020).

**Memorization in Conditional DPMs** Informative labels cause a clustering effect in the latent space of DPMs by providing contextual information that allows the encoder to better differentiate

between data samples. When data points are associated with informative labels, the encoder can map them to a latent distribution that accurately reflects their shared characteristics. This results in tighter clusters in the latent space as samples with the same informative label become more concentrated around their respective means. Consequently, the latent representations of these samples exhibit reduced variance, leading to a more structured and organized latent space. We prove these two properties under the assumption of model convergence.

Suppose we have an encoder $f_{\theta_E}(\boldsymbol{x})$ and a decoder $f_{\theta_D}(\boldsymbol{z})$. The encoder $f_{\theta_E}(\boldsymbol{x})$ maps data samples $\boldsymbol{x} \in \mathcal{D}$ to the latent distribution $\boldsymbol{z}$, which follows a normal distribution $\mathcal{N}(\boldsymbol{\mu}, \boldsymbol{\Sigma})$: $p_\theta(\boldsymbol{z}) = \mathcal{N}(\boldsymbol{\mu}, \boldsymbol{\Sigma})$. For $\boldsymbol{x}_i \in \mathcal{D}_{y=c}$, the encoder maps $\boldsymbol{x}_i$ to a latent distribution $\boldsymbol{z}_c$ subject to $\mathcal{N}(\boldsymbol{\mu_c}, \boldsymbol{\Sigma_c})$. The decoder $f_{\theta_D}(\boldsymbol{z})$ maps $\boldsymbol{z}$ back to the original data samples $\boldsymbol{x}$. $y_i$ is the label of data sample $\boldsymbol{x}_i$. Training a generative $p_\theta$ is to optimize the following:

$$\min_{\theta} - \sum_{\boldsymbol{x}_i \in \mathcal{D}} \log p_\theta\left(\boldsymbol{x}_i | y_i\right). \tag{6}$$

**Assumption 1** *Given suffix training on $\mathcal{D}$, the generative model $p_\theta$ converges to an optimal solution for Eq. (6) : $\theta^* = \arg\min_\theta - \sum_{\boldsymbol{x}_i \in \mathcal{D}} \log p_\theta(\boldsymbol{x}_i|y_i)$.*

Based on the above assumption, we can derive the following Proposition 1, with detailed proof deferred to Appendix A.3

**Proposition 1** *Let $\boldsymbol{z}$ be the latent space of generative model $p_\theta$ conditioned on an informative label $y = c$, the latent representations learned by $p_\theta$ under Assumption 1 satisfy:*

$$\sum_{\boldsymbol{z}_i \in \mathcal{D}_{y=c}} (\boldsymbol{z}_i - \boldsymbol{\mu}_c)^{\mathrm{T}} (\boldsymbol{z}_i - \boldsymbol{\mu}_c) \leq \sum_{\boldsymbol{z}_i \in \mathcal{D}_{y=c}} (\boldsymbol{z}_i - \boldsymbol{\mu})^{\mathrm{T}} (\boldsymbol{z}_i - \boldsymbol{\mu}), \tag{7}$$

$$\|\boldsymbol{\Sigma}_c\|_* \leq \|\boldsymbol{\Sigma}\|_*, \tag{8}$$

*where $\Sigma$ is the covariance matrix, $\Sigma_c$ is the covariance matrix conditioned on informative label $y = c$, $\boldsymbol{\mu}_c$ denotes the mean of the latent representations $\boldsymbol{z}$ conditioned on $y = c$ and $\boldsymbol{\mu}$ denotes the mean on the overall dataset $\mathcal{D}$.*

Proposition 1 describes two properties of the learned latent space driven by informative labels: 1) **tighter clustering** as defined in Eq. (7) and 2) **reduced variance** as defined in Eq. (8). Tighter clustering means that the data samples associated with the same informative label are more closely clustered, which allows the model to more effectively capture and memorize the specific features and patterns relevant to those labels. This proximity in the latent space enhances the model's ability to recall memorized samples during generation, as the representations are organized around distinct means. Additionally, reduced variance in these clustered representations leads to greater stability, ensuring that the model can consistently reproduce memorized outputs from the clustered latent codes. Based on this understanding, we formalize the relationship between informative labels and memorization in conditional DPMs via the following theorem.

**Theorem 2** *A generative model $p_\theta$ incurs a higher degree of pointwise memorization when conditioned on informative labels $y = c$, mathematically expressed as:*

$$\lim_{\epsilon \to 0} \frac{\mathcal{M}_{point}(\mathcal{D}_{y=c}, \theta_{y=c})}{\mathcal{M}_{point}(\mathcal{D}_{y=c}, \theta)} \leq 1 \tag{9}$$

*where $\theta_{y=c}$ denotes the parameters of the model when trained on dataset $\mathcal{D}_{y=c}$.*

The proof can be found in Appendix A.4. Theorem 2 states that any form of information labels can incur more memorization in DPMs, including conventional class labels and random labels. As shown in our proof, the informative nature of a label reduces the entropy (or uncertainty) of the data distribution conditioned on that label, leading to a more focused and memorable data representation. This explains the two empirical observations made in existing works Gu et al. (2023): 1) conditional DPMs are more prone to memorization and 2) even random labels can lead to more memorization. It also explains the findings that unconditional models do not replicate data and that text conditioning increases memorization Somepalli et al. (2022); Chen et al. (2024a).

**Memorization in Unconditional DPMs** According to Theorem 2, one could leverage informative labels to extract training data from conditional DPMs. Intuitively, the Text captions and class labels commonly used to train conditional DPMs are valid informative labels, which we call *explicit informative labels*. However, unconditional DPMs do not have explicit informative labels and thus are more difficult to extract training data from. Nevertheless, our theory indicates that the learned representation clusters by an unconditional DPM can also serve as a type of informative labels, which we call *implict information labels*. It means that if we can formulate the clustering information in the training data, we could construct implicit informative labels to help extract training data from unconditional DPMs. This motivates us to propose a new data extraction method SIDE for unconditional DPMs in the next section.

## 4 PROPOSED SIDE METHOD

In this section, we will construct implicit informative labels for unconditional DPMs, convert the implicit labels into explicit ones, and then leverage the explicit labels to extract training data.

### 4.1 CONSTRUCTING IMPLICIT INFORMATIVE LABELS

Intuitively, one could use a classifier to generate (predict) implicit labels $y_I$ during the sampling process of the diffusion model. The classifier can be a normal classifier trained on the same dataset as the diffusion model. When such a classifier is not available, random labels or representation clusters extracted by a pre-trained feature extractor (e.g., the CLIP image encoder) can also be used as the implicit labels, according to our theoretical analysis in Section 3.2. We assume an implicit label $y_I$ is learned by the target unconditional DPM, with its sampling process defined as:

$$\mathrm{d}\boldsymbol{x} = \left[f(\boldsymbol{x}, t) - g(t)^2 \left(\nabla_{\boldsymbol{x}} \log p_\theta^t(\boldsymbol{x}) + \nabla_{\boldsymbol{x}} \log p_\theta^t(y_I|\boldsymbol{x})\right)\right] \mathrm{d}t + g(t)\mathrm{d}w, \tag{10}$$

where $x$ represents the state vector, $f(x, t)$ denotes the drift coefficient, $g(t)$ is the diffusion coefficient, $\nabla_{\boldsymbol{x}} \log p_\theta^t(\boldsymbol{x})$ denotes the gradient of model $p_\theta$ given $\boldsymbol{x}$ at time $t$, and $\mathrm{d}w$ corresponds to the increment of the Wiener process.

We can use the classifier that generates the implicit labels to approximate the gradient in Eq. (10). However, a known challenge associated with neural network classifiers is their tendency towards miscalibration (Guo et al., 2017). Specifically, the classifier could be overconfident or underconfident about its predictions. To mitigate the potential impact of miscalibration on the sampling process, we introduce a hyperparameter $\lambda$ to calibrate the classifier's probability output on the diffusion path using power prior as follows:

$$p_\theta^t\left(\boldsymbol{x}|y_I\right) \propto p_\theta^{t\lambda}\left(y_I|\boldsymbol{x}\right) p_\theta^t\left(\boldsymbol{x}\right). \tag{11}$$

Then, we have:

$$\mathrm{d}\boldsymbol{x} = \left[f(\boldsymbol{x}, t) - g(t)^2 \left(\nabla_{\boldsymbol{x}} \log p_\theta^t(\boldsymbol{x}) + \lambda \nabla_{\boldsymbol{x}} \log p_\theta^t(y_I|\boldsymbol{x})\right)\right] \mathrm{d}t + g(t)\mathrm{d}w. \tag{12}$$

Note that this classifier-conditioned sampling process was initially introduced in (Dhariwal & Nichol, 2021) for a different purpose, i.e., improving sample quality with classifier guidance. Our derivation is different from (Dhariwal & Nichol, 2021). They assumed that $\int p_\theta^{t\lambda}(y|\boldsymbol{x})dy = Z$ with $Z$ being a constant. However, this assumption only holds when $\lambda = 1$, as $Z$ is explicitly dependent on the $\boldsymbol{x}_t$ (the $t$-th step of sampling image $\boldsymbol{x}$) when $\lambda \neq 1$. Our derivation solves this issue by redefining the $p_\theta^t(y|\boldsymbol{x})$ using power prior.

### 4.2 TIME-DEPENDENT CLASSIFIER

In Eq. (151), the classifier is denoted by $p_\theta^t(y|\boldsymbol{x})$ with $t$ being the timestep, implying its time-dependent nature. However, we do not have a time-dependent classifier but only a time-independent classifier. To address this problem, we propose a method named *Time-Dependent Knowledge Distillation (TDKD)* to train a time-dependent classifier. The distillation process is illustrated in Figure 2. TDKD equips the classifier with time-dependent guidance during sampling. It operates in two steps: first, the network architecture is adjusted to accommodate time-dependent inputs, and the structure of the time-dependent module and modification are illustrated in Appendix C; second, a

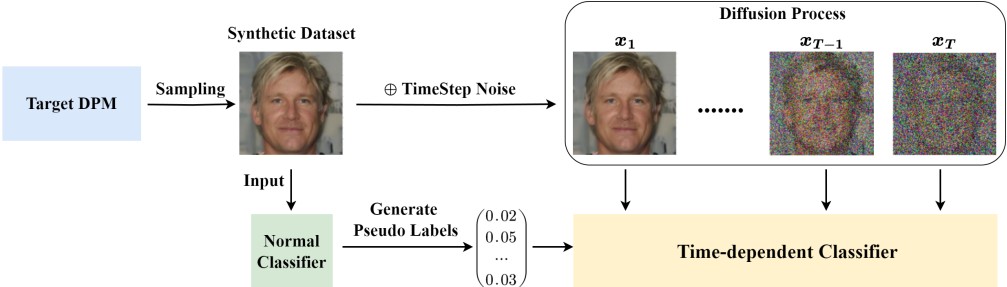

Figure 2: An illustration of our proposed Time-Dependent Knowledge Distillation (TDKD) method that trains a time-dependent classifier on a pseudo-labeled synthetic dataset.

synthetic dataset and pseudo labels are created to facilitate knowledge distillation from the normal classifier to its time-dependent counterpart.

As the original training dataset is unknown, we employ the target DPM to generate a synthetic dataset, following the generative data augmentation techniques Chen et al. (2023; 2024b). We then use the normal classifier trained on the original dataset to generate pseudo labels for the synthetic dataset. Finally, we train a time-dependent classifier on the labelled synthetic dataset. The objective of this training is defined as following:

$$\mathcal{L}_{distil} = D_{KL}\left(p_\theta(y_I|\boldsymbol{x}), p_\theta^t(y_I|\boldsymbol{x}_t)\right). \tag{13}$$

**Overall Procedure of SIDE**   With the trained time-dependent classifier $p_\theta^t(y|\boldsymbol{x}_t)$ and the target DPM, our SIDE extracts training data from the model following a conditional generation process. Assume we condition on the label $y = c$. First, we pick a set of values for $\lambda$, called $\mathcal{S}_\lambda$, to use in the SIDE attack. Then, we gather $N_G$ data samples for each value of $\lambda$ from the set. During each sampling timestep $t$, we compute the gradient $\nabla_{\boldsymbol{x}_t} CE(c, p_\theta^t(y|\boldsymbol{x}_t))$ ($CE(\cdot)$ is the cross-entropy loss), then we use the gradient and the target DPM to reverse the diffusion process. Third, we compute the similarity score for each generated image. Lastly, we evaluate the attack performance using evaluation metrics. Defer to Appendix E for SIDE's pseudocode.

## 5   EXPERIMENTS

In this section, we first describe our experimental setup, introduce the performance metrics, and then present the main evaluation results of our SIDE method. We empirically verify that the memorized images are not from the classifier. We also conduct an ablation study and hyper-parameter analysis to help understand the working mechanism of SIDE.

### 5.1   EXPERIMENTAL SETUP

We evaluate the effectiveness of our method on three datasets: 1) CelebA-HQ-Face-Identity (CelebA-HQ-FI) (Na et al., 2022) which consists of 5478 images, 2) a subset of the CelebA (CelebA-25000) (Liu et al., 2015) which contains 25,000 images and 2) CIFAR-10 containing 50,000 images. All the images are normalized to [-1,1]. We use the AdamW optimizer (Loshchilov & Hutter, 2019) with a learning rate of 1e-4 to train the time-dependent classifier. We train denoising DPMs with a discrete denoising scheduler (DDIM (Song et al., 2021)) using the HuggingFace implementation (von Platen et al., 2022). All DPMs are trained with batch size 64. We train the models for 258k steps ($\approx$ 3000 epochs) on CelebA-HQ-FI, 390k steps ($\approx$ 1000 epochs) on CelebA-25000 and 1600k steps ($\approx$ 2048 epochs) on CIFAR-10, respectively. We use ResNet34 (He et al., 2015) as the normal classifier.

### 5.2   PERFORMANCE METRICS

Determining whether a generated image is a memorized copy of a training image is difficult, as $L_p$ distances in the pixel space are ineffective. Previous research addresses this by using the 95th percentile *Self-Supervised Descriptor for Image Copy Detection (SSCD)* score for image copy

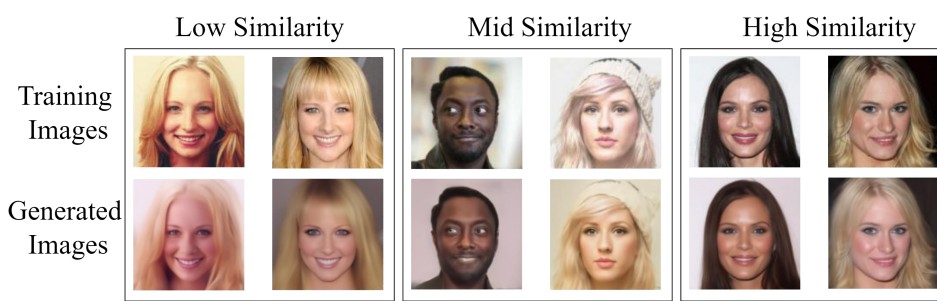

Figure 3: A comparison between the original training images (top row) and generated images (bottom row) by our SIDE method. The matches are classified into three categories: low similarity (SSCD score < 0.5), mid similarity (SSCD score between 0.5 and 0.6), and high similarity (SSCD score > 0.6). This classification highlights varying degrees of semantic resemblance among the image pairs.

detection (Somepalli et al., 2022; Gu et al., 2023). However, the 95th percentile SSCD score has three limitations: 1) it does not measure the uniqueness of memorized images; 2) it may underestimate the number of memorized samples when cut at the 95th percentile; and 3) it does not account for different types of memorization. Here, we propose two new memorization scores to solve these issues: 1) **Average Memorization Score (AMS)** and 2) **Unique Memorization Score (UMS metrics)**.

The AMS and UMS metrics are defined as follows:

$$AMS\left(\mathcal{D}_{\text{gen}}, \mathcal{D}_{\text{train}}, \alpha, \beta\right) = \frac{\sum_{\boldsymbol{x}_i \in \mathcal{D}_{\text{gen}}} \mathcal{F}\left(\boldsymbol{x}_i, \mathcal{D}_{\text{train}}, \alpha, \beta\right)}{N_G}, \tag{14}$$

$$UMS\left(\mathcal{D}_{\text{gen}}, \mathcal{D}_{\text{train}}, \alpha, \beta\right) = \frac{\left|\bigcup_{\boldsymbol{x}_i \in \mathcal{D}_{\text{gen}}} \phi\left(\boldsymbol{x}_i, \mathcal{D}_{\text{train}}, \alpha, \beta\right)\right|}{N_G}, \tag{15}$$

where $\mathcal{D}_{\text{gen}}$ is the generated dataset, $\mathcal{D}_{\text{train}}$ is the training dataset, and $\alpha$, $\beta$ are thresholds for image similarity scoring. $\mathcal{F}(\boldsymbol{x}_i, \mathcal{D}_{\text{train}}, \alpha, \beta)$ serves as a binary check for whether any training sample meets the similarity/distance criteria. $\phi(\boldsymbol{x}_i, \mathcal{D}_{\text{train}}, \alpha, \beta)$ provides the specific indices of those training samples that meet the similarity/distance criteria. Mathematically, we can represent $\mathcal{F}(x_i, \mathcal{D}_{\text{train}}, \alpha, \beta) = \mathbb{1}[\max_{x_j \in \mathcal{D}_{\text{train}}} \gamma(x_i, x_j) \geq \alpha \ \& \ \gamma(x_i, x_j) \leq \beta] \ \varphi(x_i, \mathcal{D}_{\text{train}}, \alpha, \beta) = \{j : x_j \in \mathcal{D}_{\text{train}}, \gamma(x_i, x_j) \geq \alpha \ \& \ \gamma(x_i, x_j) \leq \beta\}$. $\gamma$ represents the similarity/distance function. For low-resolution datasets, we use the normalized $L_2$ distance as $\gamma$ following Carlini et al. (2023), while for high-resolution datasets, we use the SSCD score as $\gamma$. In our experiments, the thresholds for SSCD are set to $\alpha = 0.4$ and $\beta = 0.5$ for low similarity, $\alpha = 0.5$ and $\beta = 0.6$ for mid similarity, and $\alpha = 0.6$ and $\beta = 1.0$ for high similarity. The thresholds for the normalized $L_2$ ditance (Carlini et al., 2023) are set to $\alpha = 1.5$ and $\beta = 10$ for low similarity, $\alpha = 1.4$ and $\beta = 1.5$ for mid similarity, and $\alpha = 1.35$ and $\beta = 1.4$ for high similarity.

The AMS averages the similarity scores across generated images, ensuring that memorized images are not overlooked. In contrast, the UMS quantifies distinct memorized instances by evaluating unique matches, thereby accounting for the uniqueness of the memorized images. By further categorizing the two scores into three levels—*low*, *mid*, and *high*—we obtain more comprehensive measurements for different types of memorization.

While Carlini et al. (2023); Chen et al. (2024a) introduced metrics similar to AMS and UMS, they did not account for varying levels of similarity, which is essential for assessing different types of copyright infringement, such as character or style copying (Lee et al., 2023; Sag, 2023; Sobel, 2023). Additionally, the UMS considers the number of generated images $N_G$, which was overlooked in (Carlini et al., 2023). The significance of $N_G$ lies in its non-linear impact on the UMS (see Appendix B for proof), indicating that UMS scores should not be compared across different values of $N_G$.

## 5.3 MAIN RESULTS

We compare SIDE with a random baseline and a variant of SIDE that substitutes the time-dependent classifier with a standard (time-independent) classifier. Here, "TD" refers to the time-dependent

Table 1: The AMS (%) and UMS (%) results at low (top), mid (middle), and high (bottom) levels. 'Random' denotes the baseline that generates images directly using the target unconditional DPM, while OL-TI is a variant of SIDE which is not trained using TDKD.

| Dataset | Method | Low Similarity | | Mid Similarity | | High Similarity | |
| | | AMS(%) | UMS(%) | AMS(%) | UMS(%) | AMS(%) | UMS(%) |
|---|---|---|---|---|---|---|---|
| CelebA-HQ-FI | Random | 11.656 | 2.120 | 0.596 | 0.328 | 0.044 | 0.040 |
| | OL-TI | 2.649 | 0.744 | 0.075 | 0.057 | 0.005 | 0.005 |
| | **SIDE (Ours)** | **15.172** | **2.342** | **1.115** | **0.444** | **0.054** | **0.044** |
| CelebA-25000 | Random | 5.000 | 4.240 | 0.100 | 0.100 | 0.000 | 0.000 |
| | OL-TI | 0.164 | 0.152 | 0.000 | 0.000 | 0.000 | 0.000 |
| | **SIDE (Ours)** | **8.756** | **6.940** | **0.224** | **0.212** | **0.012** | **0.012** |
| CIFAR-10 | Random | 2.470 | 1.770 | 0.910 | 0.710 | 0.510 | 0.420 |
| | OL-TI | 2.460 | 1.780 | 0.800 | 0.680 | 0.420 | 0.370 |
| | **SIDE (Ours)** | **5.325** | **2.053** | **2.495** | **0.860** | **1.770** | **0.560** |

classifier trained using our proposed TDKD method, "TI" denotes the time-independent classifier, and "OL" indicates training with the original dataset labels.

The "Random" baseline generates images directly using the target unconditional DPM, as described in (Carlini et al., 2023). We average the results across various values of $\lambda$ (defined in Eq. (151)), ranging from 5 to 9, with a detailed analysis provided in Section 5.3. It is important to note that $\lambda = 0$ corresponds to the "Random" baseline.

For each $\lambda$, including $\lambda = 0$, we generate 50,000 images for CelebA and 10,000 images for CIFAR-10 to validate our theoretical analysis and the proposed SIDE method. This effort results in one of the largest generated image datasets to date for studying the memorization of DPMs.

**Effectiveness of SIDE**    The AMS and UMS results for the three datasets are presented in Table 1. As shown, SIDE is highly effective in extracting memorized data across all similarity levels, particularly on the CelebA-25000 dataset, a task previously deemed unfeasible due to its large scale.

On CelebA-HQ-FI, SIDE increases mid-level AMS by 87% to 1.115% and UMS by 37% to 0.444%, with an average improvement of 20% across other levels. For the CelebA-25000 dataset, SIDE dramatically enhances AMS and UMS, achieving increases of 75% and 63% for low similarity and 124% and 112% for mid similarity. In the high similarity, SIDE excels at extracting memorized data.

On CIFAR-10, SIDE also outperforms the baselines across all similarity levels. For low similarity, it achieves an AMS of 5.325%, more than double Random's 2.470%, and a UMS of 2.05% compared to 1.780%. For mid similarity, SIDE reaches 2.495% AMS and 0.860% UMS, significantly higher than Random's 0.910% AMS and 0.710% UMS, respectively. In the high similarity category, SIDE achieves 1.770% AMS and 0.560% UMS, well ahead of Random's 0.510% AMS and 0.420% UMS.

**Effectiveness of TDKD**    As shown in Table 1, time-independent classifiers perform significantly worse than their time-dependent counterparts on the two high-resolution CelebA datasets, achieving only about 10% of the effectiveness of classifiers trained using our TDKD method. This discrepancy arises because time-independent classifiers can provide accurate gradients only at the final timestep, whereas time-dependent classifiers deliver accurate gradients at each timestep. However, the performance gap narrows on the CIFAR-10 dataset due to its simplicity, consisting of only 10 classes. Thus, the gradients produced by the time-independent classifier are less prone to inaccuracy, which mitigates the limitations of the time-independent classifier.

**Memorized Images Are Not From the Classifiers**    There might be a concern that the extracted images may originate from the classifier rather than the target DPM. We argue that even if a time-dependent classifier could disclose its training images, these images would still originate from the target DPM, as they can also be extracted with the time-dependent classifiers. Furthermore, Table

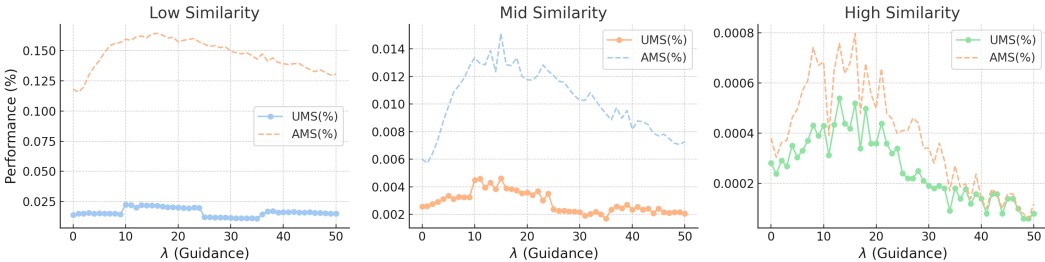

Figure 4: Hyper-parameter ($\lambda$) analysis on CelebA-HQ-FI. For high similarity, the best $\lambda$ for AMS and UMS are 16 and 13. For other similarity levels, the best $\lambda$ for AMS and UMS is 13.

1 demonstrates that employing a simple classifier reduces memorization compared to the baseline extraction method, indicating that the memorized images do not originate from the classifier.

**Impact of the Classifier on SIDE**   The classifier used to train SIDE is associated with a certain number of classes. Here, we conduct experiments to explore the relationship between the number of classes (of the classifier) and the extraction performance at a low similarity level, using 1,200 images per class with $\lambda = 5$ on the CelebA-HQ-FI dataset. As shown in Table 2 and Figure 7 (Appendix), there exists a strong positive correlation: as the number of classes increases, both AMS and UMS improve. In summary, increasing number of classes positively affects both AMS and UMS, with a stronger impact on AMS. The UMS values reported here differ significantly from Table 1. This is because here, we only generated 1,200 images per class, whereas 50,000 images per class in the previous experiment. Increasing the number of classes improves AMS more than UMS because a higher class count enables the classifier to better differentiate fine details, leading to more accurate matches at low similarity levels. UMS is less affected since it relies more on the diversity across images, which is constrained by the smaller number of images per class in this experiment.

Table 2: This table presents the results of fitting a linear model to the relationship between the number of classes and both AMS and UMS.

| Relationship | Coefficient ($\times 10^{-5}$) | Intercept | $R^2$ | Correlation Coefficient |
|---|---|---|---|---|
| #Class vs. AMS | 7.4 (positive) | 0.115 | 0.637 | 0.80 |
| #Class and UMS | 6.1 (positive) | 0.090 | 0.483 | 0.70 |

**Hyper-parameter Analysis**   Here, we test the sensitivity of SIDE to its hyper-parameter $\lambda$. We generate 50,000 images for each integer value of $\lambda$ within the range of [0, 50]. As shown in Figure 4, the memorization score increases at first, reaching its highest, then decreases as $\lambda$ increases. This can be understood from sampling SDE Eq. (151). Starting from 0, the diffusion models are unconditional. As $\lambda$ increases, the diffusion models become conditional, and according to Theorem 2, the memorization effect will be triggered. However, when $\lambda$ is too large, the generated images will overfit the classifier's decision boundaries, leading to a low diversity and ignoring the data distribution. Consequently, the memorization score decreases.

## 6 CONCLUSION

In this paper, we introduced a pointwise memorization metric to quantify memorization effects in DPMs. We provided a theoretical analysis of conditional memorization, offering a generalized definition of informative labels and clarifying that random labels can also be informative. We distinguish between *explicit labels* and *implicit labels* and propose a novel method, *SIDE*, to extract training data from unconditional diffusion models by constructing a surrogate condition. The key to this approach is training a time-dependent classifier using our *TDKD* technique. We empirically validate SIDE on subsets of the CelebA and CIFAR-10 datasets with two new memorization scores: AMS and UMS. We aim for our work to enhance the understanding of memorization mechanisms in diffusion models and inspire further methods to mitigate memorization.

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

## BROADER IMPACTS

Our work introduces a memorization metric that not only quantifies memorization effects in diffusion models but also extends to deep learning models more broadly. This contribution is significant in enhancing our understanding of when and how models memorize training data, which is critical for addressing concerns about data privacy and model robustness. By providing a theoretical framework for conditional memorization, we pave the way for developing effective memorization mitigation strategies tailored for diffusion models. These advancements can lead to the design of more secure and trustworthy AI systems, reducing the risk of potential data leakage while fostering greater accountability in the use of generative technologies. Ultimately, our findings aim to empower researchers and practitioners to create models that better respect privacy, thus benefiting the wider AI community.

## A  PROOFS

### A.1  PRELIMINARIES

If $p(x)$ and $q(x)$ are normal distributions:

$$p(\boldsymbol{x}) = \frac{1}{\sqrt{(2\pi)^d \det(\boldsymbol{\Sigma}_p)}} \exp\left\{-\frac{1}{2}(\boldsymbol{x} - \boldsymbol{\mu_p})^\top \boldsymbol{\Sigma}_p^{-1}(\boldsymbol{x} - \boldsymbol{\mu_p})\right\} \tag{16}$$

$$q(\boldsymbol{x}) = \frac{1}{\sqrt{(2\pi)^d \det(\boldsymbol{\Sigma}_q)}} \exp\left\{-\frac{1}{2}(\boldsymbol{x} - \boldsymbol{\mu_q})^\top \boldsymbol{\Sigma}_q^{-1}(\boldsymbol{x} - \boldsymbol{\mu_q})\right\} \tag{17}$$

Then we have:

$$\mathbb{E}_{\boldsymbol{x} \sim p(\boldsymbol{x})}\left[(\boldsymbol{x} - \boldsymbol{\mu}_q)^\top \boldsymbol{\Sigma}_q^{-1}(x - \boldsymbol{\mu}_q)\right] \tag{18}$$

$$= \mathrm{Tr}\left(\boldsymbol{\Sigma}_q^{-1}\boldsymbol{\Sigma}_p\right) + (\boldsymbol{\mu}_p - \boldsymbol{\mu}_q)^\top \boldsymbol{\Sigma}_q^{-1}(\boldsymbol{\mu}_p - \boldsymbol{\mu}_q) \tag{19}$$

$$\tag{20}$$

$$\mathbb{E}_{\boldsymbol{x} \sim q(\boldsymbol{x})}\left[(\boldsymbol{x} - \boldsymbol{\mu}_q)^\top \boldsymbol{\Sigma}_q^{-1}(\boldsymbol{x} - \boldsymbol{\mu}_q)\right] = d \tag{21}$$

The entropy of $p(x)$:

$$H_p(\boldsymbol{x}) = \mathbb{E}_{\boldsymbol{x} \sim p(\boldsymbol{x})}[-\log p(x)] = \frac{n}{2}(1 + \log 2\pi) + \frac{1}{2}\log\det(\boldsymbol{\Sigma}_p) \tag{22}$$

The KL divergence between the two distributions is:

$$D_{KL}(p(\boldsymbol{x})\|q(\boldsymbol{x})) \tag{23}$$

$$= \frac{1}{2}\left[(\boldsymbol{\mu}_p - \boldsymbol{\mu}_q)^\top \boldsymbol{\Sigma}_q^{-1}(\boldsymbol{\mu}_p - \boldsymbol{\mu}_q) - \log\det\left(\boldsymbol{\Sigma}_q^{-1}\boldsymbol{\Sigma}_p\right) + \mathrm{Tr}\left(\boldsymbol{\Sigma}_q^{-1}\boldsymbol{\Sigma}_p\right) - d\right] \tag{24}$$

### A.2  PROOF FOR THEOREM 1

We begin by assuming that we have an encoder $f_{\theta_E}(\boldsymbol{x})$ and a decoder $f_{\theta_D}(\boldsymbol{z})$. The encoder $f_{\theta_E}(\boldsymbol{x})$ maps the data samples $\boldsymbol{x}$ into the latent distribution $\boldsymbol{z}$, which is modeled by a normal distribution $N(\boldsymbol{\mu}, \boldsymbol{\Sigma})$, where $\boldsymbol{z} \in \mathbb{R}^d$ is the latent space of dimension $d$. The decoder $f_{\theta_D}(\boldsymbol{z})$ maps the latent variables $\boldsymbol{z}$ back to the original data samples $\boldsymbol{x}$. This structure forms the basis of many variational autoencoder (VAE) frameworks, where we aim to optimize the relationship between $\boldsymbol{x}$ and $\boldsymbol{z}$ through probabilistic modeling.

**Transformation of Probability Distributions**  Based on the transformation of probability density functions (PDF) and the method of change of variables for multiple integrals, we can express the likelihoods as follows:

1. **Conditional Probability of $x$ given $y = c$:**

$$p_\theta\left(x \mid y = c\right) = p_\theta\left(z \mid y = c\right) \det\left(\frac{\partial z}{\partial x}\right) \tag{25}$$

$$= p_\theta\left(z \mid y = c\right) \det\left(\frac{\partial f_{\theta_E}(x)}{\partial x}\right) \tag{26}$$

Here, $p_\theta\left(x \mid y = c\right)$ is the probability of the data $x$ conditioned on the label $y = c$, which depends on the latent variable $z$. The term $\det\left(\frac{\partial f_{\theta_E}(x)}{\partial x}\right)$ is the determinant of the Jacobian matrix of the encoder function $f_{\theta_E}(x)$, which accounts for the change of variables from $x$ to $z$.

2. **Marginal Probability of $x$:**

$$p_\theta\left(x\right) = p_\theta\left(z\right) \det\left(\frac{\partial z}{\partial x}\right) \tag{27}$$

$$= p_\theta\left(z\right) \det\left(\frac{\partial f_{\theta_E}(x)}{\partial x}\right) \tag{28}$$

This expression captures the marginal distribution of $x$, which is obtained by marginalizing over the latent variable $z$.

3. **Volume Element Change:**

$$dx = \det\left(\frac{\partial x}{\partial z}\right) dz \tag{29}$$

$$= \det\left(\frac{\partial f_{\theta_D}(z)}{\partial z}\right) dz \tag{30}$$

$$= \det\left(\frac{\partial x}{\partial f_{\theta_E}(x)}\right) dz \tag{31}$$

This represents the volume element transformation between the latent space $z$ and the data space $x$. The determinant of the Jacobian of the decoder $f_{\theta_D}(z)$ relates the volume elements in $z$ and $x$.

**Objective Function and Change of Variables**   Given the above transformations, we can rewrite the memorization objective $\mathcal{M}_{point}(\mathcal{D}; \theta)$ as follows:

1. **Original Memorization Objective:**

$$\mathcal{M}_{point}(\mathcal{D}; \theta) = \sum_{x_i \in \mathcal{D}} \int p_\theta(x) \log \frac{p_\theta(x)}{q(x, x_i, \epsilon)} \, dx \tag{32}$$

This measures the difference between the model distribution $p_\theta(x)$ and a perturbed distribution $q(x, x_i, \epsilon)$ at each point $x_i$ in the dataset $\mathcal{D}$.

2. **Transformed Memorization Objective in Latent Space:**

$$\mathcal{M}_{semantic}(\mathcal{D}; \theta) = \sum_{z_i \in \mathcal{D}} \int p_\theta(z) \log \frac{p_\theta(z)}{q(z, z_i, \epsilon)} \, dz \tag{33}$$

By applying the change of variables, we transform the objective into the latent space, where the same logic applies, but now the integration is over the latent variables $z$ instead of the data space $x$.

**Monotonicity Derivation**   We now derive the monotonicity of the memorization objective with respect to the content-related part $\mathcal{M}_{semantic}(\mathcal{D}; \theta)$. Specifically, we want to show that the objective increases monotonically with $\mathcal{M}_{semantic}$:

1. **Partial Derivative of the Objective:**

$$\frac{\partial \mathcal{M}(\mathcal{D}; \theta)}{\partial \mathcal{M}_{semantic}(\mathcal{D}; \theta)} = \frac{\partial}{\partial \mathcal{M}_{semantic}(\mathcal{D}; \theta)} \left( \sum_{x_i \in \mathcal{D}} \int p_\theta(x) \log \frac{p_\theta(x)}{q(x, x_i, \epsilon)} \, dz \right) \tag{34}$$

2. **Expanding the Derivatives:**

$$= \frac{\partial}{\partial \mathcal{M}_{semantic}(\mathcal{D};\theta)} \left( |D|\mathcal{H}(p_\theta) + \sum_{\boldsymbol{z}_i \in \mathcal{D}} \int p_\theta(\boldsymbol{z}) \frac{(\boldsymbol{z}-\boldsymbol{z}_i)^T (\boldsymbol{z}-\boldsymbol{z}_i)}{2\epsilon} d\boldsymbol{z} + \frac{d}{2}\log 2\pi\epsilon \right) \quad (35)$$

3. **Applying the Chain Rule:**

$$= \frac{\partial |D|\mathcal{H}(p_\theta)}{\partial \mathcal{M}_{semantic}} + \frac{\partial}{\partial \mathcal{M}_{semantic}} \left( \sum_{\boldsymbol{z}_i \in \mathcal{D}} \int p_\theta(\boldsymbol{z}) \frac{(\boldsymbol{z}-\boldsymbol{z}_i)^T (\boldsymbol{z}-\boldsymbol{z}_i)}{2\epsilon} d\boldsymbol{z} \right) + \frac{\partial}{\partial \mathcal{M}_{semantic}} \left( \frac{d}{2}\log 2\pi\epsilon \right)$$
$$(36)$$

4. **Evaluating Each Term:**

$$= \frac{\text{Tr}(\Sigma_{p_\theta}^{-1})}{2} + \frac{\partial}{\partial \mathcal{M}_{semantic}} \left( \frac{\sum_{\boldsymbol{z}_i \in \mathcal{D}} \int p_\theta(\boldsymbol{z}) (\boldsymbol{z}-\boldsymbol{z}_i)^T (\boldsymbol{z}-\boldsymbol{z}_i) d\boldsymbol{z}}{2\epsilon} \right) + 0 \quad (37)$$

5. **Final Form:**

$$= \frac{1}{2\epsilon} + \frac{\text{Tr}(\Sigma_{p_\theta}^{-1})}{2} \quad (38)$$

### A.2.1 DERIVE $\frac{\partial \mathcal{H}(p_\theta)}{\partial \mathcal{M}_{semantic}}$

We are given:

- A multivariate normal distribution $p_\theta(\boldsymbol{z}) = \mathcal{N}(\boldsymbol{\mu}, \Sigma_{p_\theta})$, where $\boldsymbol{\mu}$ is the mean vector and $\Sigma_{p_\theta}$ is the covariance matrix.
- The entropy of this distribution is:

$$H(p_\theta) = \frac{1}{2} \ln \left( (2\pi e)^d |\Sigma_{p_\theta}| \right) = \frac{d}{2} \ln(2\pi e) + \frac{1}{2} \ln |\Sigma_{p_\theta}| \quad (39)$$

- The sum expression:

$$\mathcal{M}_{semantic} = \sum_{\boldsymbol{z}_i \in \mathcal{D}} \mathbb{E}_{p_\theta} \left[ \|\boldsymbol{z}-\boldsymbol{z}_i\|^2 \right] = |\mathcal{D}| \, \text{Tr}(\Sigma_{p_\theta}) + \sum_{i=1}^{|\mathcal{D}|} \|\boldsymbol{\mu}-\boldsymbol{z}_i\|^2 \quad (40)$$

    where $|\mathcal{D}|$ is the number of data points in $\mathcal{D}$.

Our goal is to find $\frac{\partial H(p_\theta)}{\partial \mathcal{M}_{semantic}}$.

---

**Entropy of a Multivariate Normal Distribution** The entropy of a multivariate normal distribution $p_\theta(\boldsymbol{z}) = \mathcal{N}(\boldsymbol{\mu}, \Sigma_{p_\theta})$ is given by:

$$H(p_\theta) = -\int p_\theta(\boldsymbol{z}) \ln p_\theta(\boldsymbol{z}) \, d\boldsymbol{z} \quad (41)$$

$$= \frac{1}{2} \ln \left( (2\pi e)^d |\Sigma_{p_\theta}| \right) \quad (42)$$

Where:

- $d$ is the dimensionality of the vector $\boldsymbol{z}$.
- $|\Sigma_{p_\theta}|$ denotes the determinant of the covariance matrix $\Sigma_{p_\theta}$.

Thus, we can write:

$$H(p_\theta) = \frac{d}{2} \ln(2\pi e) + \frac{1}{2} \ln |\Sigma_{p_\theta}| \quad (43)$$

**Computing the Sum $\mathcal{M}_{semantic}$**    We have:

$$\mathcal{M}_{semantic} = \sum_{\boldsymbol{z}_i \in \mathcal{D}} \mathbb{E}_{p_\theta} \left[ \|\boldsymbol{z} - \boldsymbol{z}_i\|^2 \right] \tag{44}$$

First, compute $\mathbb{E}_{p_\theta} \left[ \|\boldsymbol{z} - \boldsymbol{z}_i\|^2 \right]$ for each $\boldsymbol{z}_i$:

**Expanding the Squared Norm**

$$\|\boldsymbol{z} - \boldsymbol{z}_i\|^2 = (\boldsymbol{z} - \boldsymbol{z}_i)^\top (\boldsymbol{z} - \boldsymbol{z}_i) \tag{45}$$
$$= \boldsymbol{z}^\top \boldsymbol{z} - 2\boldsymbol{z}^\top \boldsymbol{z}_i + \boldsymbol{z}_i^\top \boldsymbol{z}_i \tag{46}$$

**Taking the Expectation**

$$\mathbb{E}_{p_\theta} \left[ \|\boldsymbol{z} - \boldsymbol{z}_i\|^2 \right] = \mathbb{E}_{p_\theta} \left[ \boldsymbol{z}^\top \boldsymbol{z} \right] - 2\boldsymbol{z}_i^\top \mathbb{E}_{p_\theta} \left[ \boldsymbol{z} \right] + \boldsymbol{z}_i^\top \boldsymbol{z}_i \tag{47}$$
$$= \mathrm{Tr} \left( \mathbb{E}_{p_\theta} \left[ \boldsymbol{z} \boldsymbol{z}^\top \right] \right) - 2\boldsymbol{z}_i^\top \boldsymbol{\mu} + \|\boldsymbol{z}_i\|^2 \tag{48}$$

**Computing the Expectations**

- **First Term:** The second moment of $\boldsymbol{z}$:

$$\mathbb{E}_{p_\theta} \left[ \boldsymbol{z} \boldsymbol{z}^\top \right] = \Sigma_{p_\theta} + \boldsymbol{\mu} \boldsymbol{\mu}^\top \tag{49}$$

  Therefore:

$$\mathrm{Tr} \left( \mathbb{E}_{p_\theta} \left[ \boldsymbol{z} \boldsymbol{z}^\top \right] \right) = \mathrm{Tr}(\Sigma_{p_\theta}) + \mathrm{Tr}(\boldsymbol{\mu} \boldsymbol{\mu}^\top) = \mathrm{Tr}(\Sigma_{p_\theta}) + \|\boldsymbol{\mu}\|^2 \tag{50}$$

- **Second Term:** The mean of $\boldsymbol{z}$:

$$\mathbb{E}_{p_\theta} \left[ \boldsymbol{z} \right] = \boldsymbol{\mu} \tag{51}$$

- **Third Term:** Constant term involving $\boldsymbol{z}_i$:

$$\|\boldsymbol{z}_i\|^2 = \boldsymbol{z}_i^\top \boldsymbol{z}_i \tag{52}$$

**Combining the Terms**    Substitute back into the expectation:

$$\mathbb{E}_{p_\theta} \left[ \|\boldsymbol{z} - \boldsymbol{z}_i\|^2 \right] = \left( \mathrm{Tr}(\Sigma_{p_\theta}) + \|\boldsymbol{\mu}\|^2 \right) - 2\boldsymbol{z}_i^\top \boldsymbol{\mu} + \|\boldsymbol{z}_i\|^2 \tag{53}$$
$$= \mathrm{Tr}(\Sigma_{p_\theta}) + \left( \|\boldsymbol{\mu}\|^2 - 2\boldsymbol{\mu}^\top \boldsymbol{z}_i + \|\boldsymbol{z}_i\|^2 \right) \tag{54}$$
$$= \mathrm{Tr}(\Sigma_{p_\theta}) + \|\boldsymbol{\mu} - \boldsymbol{z}_i\|^2 \tag{55}$$

Thus, for each $\boldsymbol{z}_i$:

$$\mathbb{E}_{p_\theta} \left[ \|\boldsymbol{z} - \boldsymbol{z}_i\|^2 \right] = \mathrm{Tr}(\Sigma_{p_\theta}) + \|\boldsymbol{\mu} - \boldsymbol{z}_i\|^2 \tag{56}$$

**Summing Over All Data Points**    The total sum $\mathcal{M}_{semantic}$ becomes:

$$\mathcal{M}_{semantic} = \sum_{i=1}^{|\mathcal{D}|} \left( \mathrm{Tr}(\Sigma_{p_\theta}) + \|\boldsymbol{\mu} - \boldsymbol{z}_i\|^2 \right) \tag{57}$$

$$= |\mathcal{D}| \, \mathrm{Tr}(\Sigma_{p_\theta}) + \sum_{i=1}^{|\mathcal{D}|} \|\boldsymbol{\mu} - \boldsymbol{z}_i\|^2 \tag{58}$$

Let's define the sample variance $\mathrm{Var}_{\text{data}}$:

$$\text{Var}_{\text{data}} = \frac{1}{|\mathcal{D}|} \sum_{i=1}^{|\mathcal{D}|} \|\boldsymbol{\mu} - \boldsymbol{z}_i\|^2 \tag{59}$$

Then, $\mathcal{M}_{semantic}$ can be expressed as:

$$\mathcal{M}_{semantic} = |\mathcal{D}| \operatorname{Tr}(\Sigma_{p_\theta}) + |\mathcal{D}|\text{Var}_{\text{data}} = |\mathcal{D}| \left( \operatorname{Tr}(\Sigma_{p_\theta}) + \text{Var}_{\text{data}} \right) \tag{60}$$

**Attempting to Relate $H(p_\theta)$ and $\mathcal{M}_{semantic}$**  Our challenge is to express $H(p_\theta)$ as a function of $\mathcal{M}_{semantic}$ so that we can compute $\frac{\partial H(p_\theta)}{\partial \mathcal{M}_{semantic}}$. However, we face a difficulty:

- The entropy $H(p_\theta)$ depends on $\ln |\Sigma_{p_\theta}|$.
- The sum $\mathcal{M}_{semantic}$ depends on $\operatorname{Tr}(\Sigma_{p_\theta})$.

For a general covariance matrix $\Sigma_{p_\theta}$, there is no direct algebraic relationship between $\operatorname{Tr}(\Sigma_{p_\theta})$ and $\ln |\Sigma_{p_\theta}|$. Therefore, we need to explore an alternative method.

**Computing Derivatives with Respect to $\Sigma_{p_\theta}$**  The entropy $H(p_\theta)$ is given by:

$$H(p_\theta) = \frac{d}{2} \ln(2\pi e) + \frac{1}{2} \ln |\Sigma_{p_\theta}| \tag{61}$$

To find the derivative of $H(p_\theta)$ with respect to $\Sigma_{p_\theta}$, we proceed as follows:

$$\frac{\partial H(p_\theta)}{\partial \Sigma_{p_\theta}} = \frac{\partial}{\partial \Sigma_{p_\theta}} \left( \frac{d}{2} \ln(2\pi e) + \frac{1}{2} \ln |\Sigma_{p_\theta}| \right) \tag{62}$$

$$= \frac{1}{2} \frac{\partial}{\partial \Sigma_{p_\theta}} \ln |\Sigma_{p_\theta}| \tag{63}$$

$$= \frac{1}{2} \Sigma_{p_\theta}^{-1} \tag{64}$$

Recall that:

$$\mathcal{M}_{semantic} = |\mathcal{D}| \left( \operatorname{Tr}(\Sigma_{p_\theta}) + \text{Var}_{\text{data}} \right) \tag{65}$$

Since $\text{Var}_{\text{data}}$ does not depend on $\Sigma_{p_\theta}$, the derivative of $\mathcal{M}_{semantic}$ with respect to $\Sigma_{p_\theta}$ is:

$$\frac{\partial \mathcal{M}_{semantic}}{\partial \Sigma_{p_\theta}} = |\mathcal{D}| \frac{\partial}{\partial \Sigma_{p_\theta}} \operatorname{Tr}(\Sigma_{p_\theta}) \tag{66}$$

**Computing $\frac{\partial H(p_\theta)}{\partial \mathcal{M}_{semantic}}$ Using the Chain Rule**  Using the chain rule for derivatives:

$$\frac{\partial H(p_\theta)}{\partial \mathcal{M}_{semantic}} = \operatorname{Tr} \left( \frac{\partial H(p_\theta)}{\partial \Sigma_{p_\theta}} \cdot \left( \frac{\partial \mathcal{M}_{semantic}}{\partial \Sigma_{p_\theta}} \right)^{-1} \right) \tag{67}$$

**Computing** $\left(\frac{\partial \mathcal{M}_{semantic}}{\partial \Sigma_{p_\theta}}\right)^{-1}$   Since $\frac{\partial \mathcal{M}_{semantic}}{\partial \Sigma_{p_\theta}} = |\mathcal{D}|I$, its inverse is:

$$\left(\frac{\partial \mathcal{M}_{semantic}}{\partial \Sigma_{p_\theta}}\right)^{-1} = \frac{1}{|\mathcal{D}|}I \tag{68}$$

Combine together, we can derive

$$\frac{\partial H(p_\theta)}{\partial \mathcal{M}_{semantic}} = \frac{\text{Tr}(\Sigma_{p_\theta}^{-1})}{2|\mathcal{D}|} \tag{69}$$

Thus, we conclude that the memorization objective increases monotonically with respect to the memorization metric.

### A.3   PROOF FOR PROPOSITION 1

#### A.3.1   PROOF FOR EQUATION 8

**Covariance Definitions:**

$$\Sigma = \text{Cov}(Z), \quad \Sigma_c = \text{Cov}(Z \mid Y) \tag{70}$$

Here:

- $\Sigma$ is the overall covariance matrix of $Z$,
- $\Sigma_c$ is the conditional covariance matrix of $Z$ given $Y$.

The goal is to prove that the trace of the conditional covariance matrix is less than or equal to the trace of the overall covariance matrix, i.e.,

$$\text{Tr}(\Sigma_c) \leq \text{Tr}(\Sigma) \tag{71}$$

for all realizations of $Y$.

**Proof of Trace Inequality**

$$\Sigma = \text{Cov}(Z), \quad \Sigma_c = \text{Cov}(Z \mid Y) \tag{72}$$

**Here:**

- $\Sigma$ is the overall covariance matrix of the random vector $Z \in \mathbb{R}^n$,
- $\Sigma_c$ is the conditional covariance matrix of $Z$ given $Y$.

**Goal** Prove that:

$$\text{Tr}(\Sigma_c) \leq \text{Tr}(\Sigma) \tag{73}$$

for all realizations of $Y$.

We begin by expressing the overall covariance matrix $\Sigma$ in terms of the conditional covariance matrix $\Sigma_c$ and the covariance of the conditional expectation of $Z$ given $Y$.

$$\Sigma = \text{Cov}(Z) \tag{74}$$
$$= \mathbb{E}\left[(Z - \mathbb{E}[Z])(Z - \mathbb{E}[Z])^T\right] \tag{75}$$
$$= \mathbb{E}\left[(Z - \mathbb{E}[Z \mid Y] + \mathbb{E}[Z \mid Y] - \mathbb{E}[Z])(Z - \mathbb{E}[Z \mid Y] + \mathbb{E}[Z \mid Y] - \mathbb{E}[Z])^T\right] \tag{76}$$

**Expanding the Product Inside the Expectation**   We expand the product inside the expectation:

$$\Sigma = \mathbb{E}\Bigg[ \underbrace{(Z - \mathbb{E}[Z \mid Y])(Z - \mathbb{E}[Z \mid Y])^T}_{\text{Term 1}}$$

$$+ \underbrace{(Z - \mathbb{E}[Z \mid Y])(\mathbb{E}[Z \mid Y] - \mathbb{E}[Z])^T}_{\text{Term 2}}$$

$$+ \underbrace{(\mathbb{E}[Z \mid Y] - \mathbb{E}[Z])(Z - \mathbb{E}[Z \mid Y])^T}_{\text{Term 3}}$$

$$+ \underbrace{(\mathbb{E}[Z \mid Y] - \mathbb{E}[Z])(\mathbb{E}[Z \mid Y] - \mathbb{E}[Z])^T}_{\text{Term 4}} \Bigg] \tag{77}$$

**Analyzing Each Term**

**Term 1:**

$$\mathbb{E}\left[(Z - \mathbb{E}[Z \mid Y])(Z - \mathbb{E}[Z \mid Y])^T\right] = \text{Cov}(Z \mid Y) = \Sigma_c \tag{78}$$

**Term 2:**

$$\mathbb{E}\left[(Z - \mathbb{E}[Z \mid Y])(\mathbb{E}[Z \mid Y] - \mathbb{E}[Z])^T\right] \tag{79}$$

To evaluate Term 2, we condition on $Y$:

$$\mathbb{E}\left[(Z - \mathbb{E}[Z \mid Y])(\mathbb{E}[Z \mid Y] - \mathbb{E}[Z])^T\right] = \mathbb{E}\left[\mathbb{E}\left[(Z - \mathbb{E}[Z \mid Y])(\mathbb{E}[Z \mid Y] - \mathbb{E}[Z])^T \mid Y\right]\right] \tag{80}$$

Inside the inner expectation, $\mathbb{E}[Z \mid Y] - \mathbb{E}[Z]$ is treated as a constant with respect to $Z$, so:

$$\mathbb{E}\left[(Z - \mathbb{E}[Z \mid Y])(\mathbb{E}[Z \mid Y] - \mathbb{E}[Z])^T \mid Y\right] \tag{81}$$

$$= (\mathbb{E}[Z \mid Y] - \mathbb{E}[Z]) \, \mathbb{E}\left[Z - \mathbb{E}[Z \mid Y] \mid Y\right] \tag{82}$$

$$= (\mathbb{E}[Z \mid Y] - \mathbb{E}[Z]) \cdot 0 = 0 \tag{83}$$

Thus:

$$\mathbb{E}\left[(Z - \mathbb{E}[Z \mid Y])(\mathbb{E}[Z \mid Y] - \mathbb{E}[Z])^T\right] = 0 \tag{84}$$

**Term 3:**

$$\mathbb{E}\left[(\mathbb{E}[Z \mid Y] - \mathbb{E}[Z])(Z - \mathbb{E}[Z \mid Y])^T\right] \tag{85}$$

Similarly, we condition on $Y$:

$$\mathbb{E}\left[(\mathbb{E}[Z \mid Y] - \mathbb{E}[Z])(Z - \mathbb{E}[Z \mid Y])^T\right] \tag{86}$$

$$= \mathbb{E}\left[(\mathbb{E}[Z \mid Y] - \mathbb{E}[Z])\mathbb{E}\left[(Z - \mathbb{E}[Z \mid Y])^T \mid Y\right]\right] \tag{87}$$

$$= \mathbb{E}\left[(\mathbb{E}[Z \mid Y] - \mathbb{E}[Z]) \cdot 0^T\right] \tag{88}$$

$$= 0 \tag{89}$$

**Term 4:**

$$\mathbb{E}\left[(\mathbb{E}[Z \mid Y] - \mathbb{E}[Z])(\mathbb{E}[Z \mid Y] - \mathbb{E}[Z])^T\right] = \text{Cov}\left(\mathbb{E}[Z \mid Y]\right) = \text{Cov}(\mathbb{E}[Z \mid Y]) \tag{90}$$

**Combining All Terms**   Putting all the terms together:

$$\Sigma = \Sigma_c + 0 + 0 + \text{Cov}(\mathbb{E}[Z \mid Y]) \tag{91}$$

$$= \Sigma_c + \text{Cov}(\mathbb{E}[Z \mid Y]) \tag{92}$$

**Taking the Trace**    Taking the trace on both sides of the covariance decomposition:

$$\text{Tr}(\Sigma) = \text{Tr}\left(\Sigma_c\right) + \text{Tr}\left(\text{Cov}(Z \mid Y)\right) \tag{93}$$
$$= \left[\text{Tr}(\Sigma_c)\right] + \text{Tr}\left(\text{Cov}(\mathbb{E}[Z \mid Y])\right) \tag{94}$$

Since the trace of a covariance matrix is non-negative:

$$\text{Tr}\left(\text{Cov}(\mathbb{E}[Z \mid Y])\right) \geq 0 \tag{95}$$

Thus:

$$\text{Tr}(\Sigma_c) \leq \text{Tr}(\Sigma) \tag{96}$$

**Conclusion**    We have proven that the trace of the conditional covariance matrix $\Sigma_c$ is less than or equal to the trace of the overall covariance matrix $\Sigma$. Formally,

$$\text{Tr}(\Sigma_c) \leq \text{Tr}(\Sigma) \tag{97}$$

This result is a multivariate generalization of the variance decomposition, showing that the expected conditional variability of $Z$ given $Y$ does not exceed the overall variability of $Z$.

### A.3.2    PROOF FOR EQUATION 7

**Centering and Decomposing**    By the definition of variance, we can express the sum of squared distances for the conditional mean $\boldsymbol{\mu}_c$ and overall mean $\boldsymbol{\mu}$:

$$\sum_{\boldsymbol{z}_i \in \mathcal{D}_{y=c}} (\boldsymbol{z}_i - \boldsymbol{\mu})^\text{T}(\boldsymbol{z}_i - \boldsymbol{\mu}) \tag{98}$$

$$\sum_{\boldsymbol{z}_i \in \mathcal{D}_{y=c}} (\boldsymbol{z}_i - \boldsymbol{\mu})^\text{T}(\boldsymbol{z}_i - \boldsymbol{\mu}) = \sum_{\boldsymbol{z}_i \in \mathcal{D}_{y=c}} \left((\boldsymbol{z}_i - \boldsymbol{\mu}_c + \boldsymbol{\mu}_c - \boldsymbol{\mu})^\text{T}(\boldsymbol{z}_i - \boldsymbol{\mu}_c + \boldsymbol{\mu}_c - \boldsymbol{\mu})\right) \tag{99}$$

$$= \sum_{\boldsymbol{z}_i \in \mathcal{D}_{y=c}} \Big[(\boldsymbol{z}_i - \boldsymbol{\mu}_c)^\text{T}(\boldsymbol{z}_i - \boldsymbol{\mu}_c)$$
$$+ (\boldsymbol{z}_i - \boldsymbol{\mu}_c)^\text{T}(\boldsymbol{\mu}_c - \boldsymbol{\mu})$$
$$+ (\boldsymbol{\mu}_c - \boldsymbol{\mu})^\text{T}(\boldsymbol{z}_i - \boldsymbol{\mu}_c)$$
$$+ (\boldsymbol{\mu}_c - \boldsymbol{\mu})^\text{T}(\boldsymbol{\mu}_c - \boldsymbol{\mu})\Big] \tag{100}$$

$$= \sum_{\boldsymbol{z}_i \in \mathcal{D}_{y=c}} (\boldsymbol{z}_i - \boldsymbol{\mu}_c)^\text{T}(\boldsymbol{z}_i - \boldsymbol{\mu}_c)$$
$$+ \sum_{\boldsymbol{z}_i \in \mathcal{D}_{y=c}} (\boldsymbol{z}_i - \boldsymbol{\mu}_c)^\text{T}(\boldsymbol{\mu}_c - \boldsymbol{\mu})$$
$$+ \sum_{\boldsymbol{z}_i \in \mathcal{D}_{y=c}} (\boldsymbol{\mu}_c - \boldsymbol{\mu})^\text{T}(\boldsymbol{z}_i - \boldsymbol{\mu}_c)$$
$$+ \sum_{\boldsymbol{z}_i \in \mathcal{D}_{y=c}} (\boldsymbol{\mu}_c - \boldsymbol{\mu})^\text{T}(\boldsymbol{\mu}_c - \boldsymbol{\mu}) \tag{101}$$

$$= \sum_{\boldsymbol{z}_i \in \mathcal{D}_{y=c}} (\boldsymbol{z}_i - \boldsymbol{\mu}_c)^\text{T}(\boldsymbol{z}_i - \boldsymbol{\mu}_c)$$
$$+ (\boldsymbol{\mu}_c - \boldsymbol{\mu})^\text{T}(\boldsymbol{\mu}_c - \boldsymbol{\mu}) \sum_{\boldsymbol{z}_i \in \mathcal{D}_{y=c}} 1 \tag{102}$$

$$= \sum_{\boldsymbol{z}_i \in \mathcal{D}_{y=c}} (\boldsymbol{z}_i - \boldsymbol{\mu}_c)^\text{T}(\boldsymbol{z}_i - \boldsymbol{\mu}_c) + |\mathcal{D}_{y=c}|(\boldsymbol{\mu}_c - \boldsymbol{\mu})^\text{T}(\boldsymbol{\mu}_c - \boldsymbol{\mu}) \tag{103}$$

Expanding this expression, we have:

$$= \sum_{\boldsymbol{z}_i \in \mathcal{D}_{y=c}} \left((\boldsymbol{z}_i - \boldsymbol{\mu}_c)^\text{T}(\boldsymbol{z}_i - \boldsymbol{\mu}_c) + 2(\boldsymbol{z}_i - \boldsymbol{\mu}_c)^\text{T}(\boldsymbol{\mu}_c - \boldsymbol{\mu}) + (\boldsymbol{\mu}_c - \boldsymbol{\mu})^\text{T}(\boldsymbol{\mu}_c - \boldsymbol{\mu})\right). \tag{104}$$

**Step 2: Simplifying the Inequality** The term $(\boldsymbol{\mu}_c - \boldsymbol{\mu})^{\mathrm{T}}(\boldsymbol{\mu}_c - \boldsymbol{\mu})$ is a constant for the conditional samples, and the term $2(\boldsymbol{z}_i - \boldsymbol{\mu}_c)^{\mathrm{T}}(\boldsymbol{\mu}_c - \boldsymbol{\mu})$ sums to zero when averaged over the samples in $\mathcal{D}_{y=c}$ due to the definition of $\boldsymbol{\mu}_c$.

Thus, we have:

$$\sum_{\boldsymbol{z}_i \in \mathcal{D}_{y=c}} (\boldsymbol{z}_i - \boldsymbol{\mu})^{\mathrm{T}}(\boldsymbol{z}_i - \boldsymbol{\mu}) = \sum_{\boldsymbol{z}_i \in \mathcal{D}_{y=c}} (\boldsymbol{z}_i - \boldsymbol{\mu}_c)^{\mathrm{T}}(\boldsymbol{z}_i - \boldsymbol{\mu}_c) + |\mathcal{D}_{y=c}|(\boldsymbol{\mu}_c - \boldsymbol{\mu})^{\mathrm{T}}(\boldsymbol{\mu}_c - \boldsymbol{\mu}),$$

$$(105)$$

where $|\mathcal{D}_{y=c}|$ is the number of samples with label $y = c$.

**Step 3: Establishing the Inequality** Since the variance (sum of squared distances) from the conditional mean will always be less than or equal to the variance from the overall mean, we conclude:

$$\sum_{\boldsymbol{z}_i \in \mathcal{D}_{y=c}} (\boldsymbol{z}_i - \boldsymbol{\mu}_c)^{\mathrm{T}}(\boldsymbol{z}_i - \boldsymbol{\mu}_c) \leq \sum_{\boldsymbol{z}_i \in \mathcal{D}_{y=c}} (\boldsymbol{z}_i - \boldsymbol{\mu})^{\mathrm{T}}(\boldsymbol{z}_i - \boldsymbol{\mu}). \tag{106}$$

This establishes that the latent space representation conditioned on an informative label exhibits reduced variance, confirming our initial claim.

GENERALIZED VARIANCE COMPARISON

In this section, we examine a broader comparison of variance that does not restrict the analysis to the subset $\mathcal{D}_{y=c}$ but rather considers the entire dataset $\mathcal{D}$. Namely, we prove that:

$$\sum_{\boldsymbol{z}_i \in \mathcal{D}_{y=c}} (\boldsymbol{z}_i - \boldsymbol{\mu}_c)^{T}(\boldsymbol{z}_i - \boldsymbol{\mu}_c) \leq \sum_{\boldsymbol{z}_i \in \mathcal{D}} (\boldsymbol{z}_i - \boldsymbol{\mu})^{T}(\boldsymbol{z}_i - \boldsymbol{\mu}) \tag{107}$$

For a set of data points $D$, the sum of squared deviations from the mean is given by:

$$\sum_{\boldsymbol{z}_i \in D} (\boldsymbol{z}_i - \boldsymbol{\mu})^{T}(\boldsymbol{z}_i - \boldsymbol{\mu}) \tag{108}$$

**Derivation:** First, let's write out the sums of squared deviations for the unconditional and conditional cases.

**Unconditional Sum of Squared Deviations:**

$$\sum_{\boldsymbol{z}_i \in D} (\boldsymbol{z}_i - \boldsymbol{\mu})^{T}(\boldsymbol{z}_i - \boldsymbol{\mu}) \tag{109}$$

where $D$ represents the entire dataset of $\boldsymbol{z}$ values.

**Conditional Sum of Squared Deviations:**

$$\sum_{\boldsymbol{z}_i \in D_{y=c}} (\boldsymbol{z}_i - \boldsymbol{\mu}_c)^{T}(\boldsymbol{z}_i - \boldsymbol{\mu}_c) \tag{110}$$

where $D_{y=c}$ represents the subset of data points $\boldsymbol{z}_i$ where $y = c$.

**Covariance Matrix and Sum of Squared Deviations:** The covariance matrix can be related to the sum of squared deviations. For the unconditional case:

$$\Sigma = \frac{1}{n} \sum_{\boldsymbol{z}_i \in D} (\boldsymbol{z}_i - \boldsymbol{\mu})(\boldsymbol{z}_i - \boldsymbol{\mu})^T \tag{111}$$

Taking the trace on both sides:

$$\text{tr}(\Sigma) = \frac{1}{n} \text{tr} \left( \sum_{\boldsymbol{z}_i \in D} (\boldsymbol{z}_i - \boldsymbol{\mu})(\boldsymbol{z}_i - \boldsymbol{\mu})^T \right) \tag{112}$$

Since the trace of a sum is the sum of the traces:

$$\text{tr}(\Sigma) = \frac{1}{n} \sum_{\boldsymbol{z}_i \in D} \text{tr} \left( (\boldsymbol{z}_i - \boldsymbol{\mu})(\boldsymbol{z}_i - \boldsymbol{\mu})^T \right) \tag{113}$$

The trace of the outer product of a vector with itself is the sum of squared elements of the vector:

$$\text{tr} \left( (\boldsymbol{z}_i - \boldsymbol{\mu})(\boldsymbol{z}_i - \boldsymbol{\mu})^T \right) = (\boldsymbol{z}_i - \boldsymbol{\mu})^T (\boldsymbol{z}_i - \boldsymbol{\mu}) \tag{114}$$

Therefore:

$$\text{tr}(\Sigma) = \frac{1}{n} \sum_{\boldsymbol{z}_i \in D} (\boldsymbol{z}_i - \boldsymbol{\mu})^T (\boldsymbol{z}_i - \boldsymbol{\mu}) \tag{115}$$

Similarly, for the conditional case:

$$\Sigma_c = \frac{1}{n_c} \sum_{\boldsymbol{z}_i \in D_{y=c}} (\boldsymbol{z}_i - \boldsymbol{\mu}_c)(\boldsymbol{z}_i - \boldsymbol{\mu}_c)^T \tag{116}$$

Taking the trace:

$$\text{tr}(\Sigma_c) = \frac{1}{n_c} \sum_{\boldsymbol{z}_i \in D_{y=c}} (\boldsymbol{z}_i - \boldsymbol{\mu}_c)^T (\boldsymbol{z}_i - \boldsymbol{\mu}_c) \tag{117}$$

**Inequality of Traces:** Given that conditioning on $y = c$ provides information about $\boldsymbol{z}$, it generally reduces the variance of $\boldsymbol{z}$. Mathematically, this can be expressed as:

$$\text{tr}(\Sigma_c) \leq \text{tr}(\Sigma) \tag{118}$$

In terms of sums of squared deviations:

$$\frac{1}{n_c} \sum_{\boldsymbol{z}_i \in D_{y=c}} (\boldsymbol{z}_i - \boldsymbol{\mu}_c)^T (\boldsymbol{z}_i - \boldsymbol{\mu}_c) \leq \frac{1}{n} \sum_{\boldsymbol{z}_i \in D} (\boldsymbol{z}_i - \boldsymbol{\mu})^T (\boldsymbol{z}_i - \boldsymbol{\mu}) \tag{119}$$

Multiplying both sides by their respective sample sizes $n_c$ and $n$:

$$\sum_{\boldsymbol{z}_i \in D_{y=c}} (\boldsymbol{z}_i - \boldsymbol{\mu}_c)^T(\boldsymbol{z}_i - \boldsymbol{\mu}_c) \leq \frac{n_c}{n} \sum_{\boldsymbol{z}_i \in D} (\boldsymbol{z}_i - \boldsymbol{\mu})^T(\boldsymbol{z}_i - \boldsymbol{\mu}) \tag{120}$$

Since $n_c \leq n$, this further simplifies to:

$$\sum_{\boldsymbol{z}_i \in D_{y=c}} (\boldsymbol{z}_i - \boldsymbol{\mu}_c)^T(\boldsymbol{z}_i - \boldsymbol{\mu}_c) \leq \sum_{\boldsymbol{z}_i \in D} (\boldsymbol{z}_i - \boldsymbol{\mu})^T(\boldsymbol{z}_i - \boldsymbol{\mu}) \tag{121}$$

### A.4 Proof for theorem 2

This section will detail the proof for the theorem 2.

$$\lim_{\epsilon \to 0} \frac{\sum_{\boldsymbol{x}_i \in \mathcal{D}_{y=c}} \int p_\theta(\boldsymbol{x}|y=c) \log \frac{p_\theta(\boldsymbol{x}|y=c)}{q(\boldsymbol{x},\boldsymbol{x}_i,\epsilon)}\, dx}{\sum_{\boldsymbol{x}_i \in \mathcal{D}_{y=c}} \int p_\theta(\boldsymbol{x}) \log \frac{p_\theta(\boldsymbol{x})}{q(\boldsymbol{x},\boldsymbol{x}_i,\epsilon)}\, d\boldsymbol{x}} \leq 1 \tag{122}$$

Define $\mathcal{D}^z_{y=c} = \{\boldsymbol{z}_i : f_{\theta_E}(\boldsymbol{x}_i) \in \mathcal{D}_{y=c}\}$

$\mathcal{D}_{y=c} = \{\boldsymbol{x}_i : \boldsymbol{x}_i \in \mathcal{D}, y_i = c\}$

By using the change of variable theorem, the (122) becomes:

$$\lim_{\epsilon \to 0} \frac{\sum_{\boldsymbol{x}_i \in \mathcal{D}_{y=c}} \int p_\theta(\boldsymbol{x}|y=c) \log \frac{p_\theta(\boldsymbol{x}|y=c)}{q(\boldsymbol{x},\boldsymbol{x}_i,\epsilon)}\, dx}{\sum_{\boldsymbol{x}_i \in \mathcal{D}_{y=c}} \int p_\theta(\boldsymbol{x}) \log \frac{p_\theta(\boldsymbol{x})}{q(\boldsymbol{x},\boldsymbol{x}_i,\epsilon)}\, d\boldsymbol{x}} \leq 1 \tag{123}$$

$$\Rightarrow \lim_{\epsilon \to 0} \frac{\sum_{\boldsymbol{z}_i \in \mathcal{D}^z_{y=c}} \int p_\theta(\boldsymbol{z}|y=c) \log \frac{p_\theta(\boldsymbol{z}|y=c)}{q(\boldsymbol{z},\boldsymbol{z}_i,\epsilon)}\, dz}{\sum_{\boldsymbol{z}_i \in \mathcal{D}^z_{y=c}} \int p_\theta(\boldsymbol{z}) \log \frac{p_\theta(\boldsymbol{z})}{q(\boldsymbol{z},\boldsymbol{z}_i,\epsilon)}\, dz} \leq 1 \tag{124}$$

Since $p_\theta(\boldsymbol{z}) = N(\boldsymbol{\mu}, \boldsymbol{\Sigma})$, it is reasonable to assume that its conditional distribution is also a normal distribution. Then:

$$p_\theta(\boldsymbol{z}|y=c) = N(\boldsymbol{\mu}_c, \boldsymbol{\Sigma}_c) \tag{125}$$

where $\boldsymbol{\mu}_c \in \mathbb{R}^d, \boldsymbol{\Sigma}_c \in \mathbb{R}^{d \times d}$. Moreover, because $p_\theta(\boldsymbol{z}|y=c)$ depends on label $c$, we can derive the following:

$$\sum_{\boldsymbol{z}_i \in \mathcal{D}^z_{y=c}} (\boldsymbol{z}_i - \boldsymbol{\mu}_c)^{\mathrm{T}}(\boldsymbol{z}_i - \boldsymbol{\mu}_c) \leq \sum_{\boldsymbol{z}_i \in \mathcal{D}^z_{y=c}} (\boldsymbol{z}_i - \boldsymbol{\mu})^{\mathrm{T}}(\boldsymbol{z}_i - \boldsymbol{\mu}) \tag{126}$$

where $\forall \boldsymbol{z}_i, f_{\theta_D}(\boldsymbol{z}_i) \in y_c$.

Intuitively, (126) means that the latent code of each training sample conditioned on the label $y = c$ is more centered around the learned latent space of distribution $p_\theta(\boldsymbol{z}|y=c)$ than around the distribution $p_\theta(\boldsymbol{z})$.

We now look into the KL divergence:

$$\int p_\theta(\boldsymbol{z}|y=c) \log \frac{p_\theta(\boldsymbol{z}|y=c)}{q(\boldsymbol{z}; \boldsymbol{z}_i)}\, d\boldsymbol{z} \tag{127}$$

$$= \int p_\theta(\boldsymbol{z}|y=c) \log p_\theta(\boldsymbol{z}|y=c) d\boldsymbol{z} - \int p_\theta(\boldsymbol{z}|y=c) \log q(\boldsymbol{z}; \boldsymbol{z}_i) d\boldsymbol{z} \tag{128}$$

$$= -\frac{d}{2}(1 + \log 2\pi) - \frac{1}{2} \log \det(\boldsymbol{\Sigma}_c) + \mathbb{E}_{\boldsymbol{z} \sim p_\theta(\boldsymbol{z}|y=c)}(-\log q(\boldsymbol{z}; \boldsymbol{z}_i)) \tag{129}$$

$$= \frac{1}{2}\left[\frac{(\boldsymbol{z}_i - \boldsymbol{\mu}_c)^{\mathrm{T}}(\boldsymbol{z}_i - \boldsymbol{\mu}_c)}{\epsilon} - \log \frac{\det(\boldsymbol{\Sigma}_c)}{\epsilon^d} + \frac{\mathrm{Tr}(\boldsymbol{\Sigma}_c)}{\epsilon} - d\right] \tag{130}$$

We use the SVD decomposition to decompose $\mathbf{\Sigma}_c$:

$$\mathbf{\Sigma}_c = U_c \Lambda_c U_c^{\mathrm{T}} \tag{131}$$

And:

$$\log \det \mathbf{\Sigma}_c = \log \det U_c \Lambda U_c^{\mathrm{T}} = \log |U_c||\Lambda_c||U_c^{\mathrm{T}}| = \log |\Lambda_c| \tag{132}$$

$$\mathrm{Tr}\left(\mathbf{\Sigma}_c\right) = \mathrm{Tr}\left(U_c \Lambda_c U_c^{\mathrm{T}}\right) = \mathrm{Tr}\left(\Lambda_c U_c U_c^{\mathrm{T}}\right) = \mathrm{Tr}\left(\Lambda_c\right) \tag{133}$$

Thus, (130) simplifies to:

$$\frac{1}{2}\left[\frac{(\boldsymbol{z}_i - \boldsymbol{\mu}_c)^{\top}(\boldsymbol{z}_i - \boldsymbol{\mu}_c)}{\epsilon} - \log\frac{\det\left(\Lambda_c\right)}{\epsilon^d} + \frac{\mathrm{Tr}\left(\Lambda_c\right)}{\epsilon} - d\right] \tag{134}$$

Similarly:

$$\tag{135}$$

$$\int p_\theta(\boldsymbol{z}) \log \frac{p_\theta(\boldsymbol{z})}{q(\boldsymbol{z}; \boldsymbol{z}_i)} \, d\boldsymbol{z} \tag{136}$$

$$= \frac{1}{2}\left[\frac{(\boldsymbol{z}_i - \boldsymbol{\mu})^{\top}(\boldsymbol{z}_i - \boldsymbol{\mu})}{\epsilon} - \log\frac{\det\left(\mathbf{\Sigma}_c\right)}{\epsilon^d} + \frac{\mathrm{Tr}\left(\mathbf{\Sigma}_c\right)}{\epsilon} - d\right] \tag{137}$$

$$= \frac{1}{2}\left[\frac{(\boldsymbol{z}_i - \boldsymbol{\mu})^{\top}(\boldsymbol{z}_i - \boldsymbol{\mu})}{\epsilon} - \log\frac{\det\left(\Lambda\right)}{\epsilon^d} + \frac{\mathrm{Tr}\left(\Lambda\right)}{\epsilon} - d\right] \tag{138}$$

where

$$\mathbf{\Sigma} = U \Lambda U^{\mathrm{T}} \tag{139}$$

According to the Eq. (8), the nuclear norm of the two covariance matrices differs. Specifically:

$$\|\mathbf{\Sigma}_c\|_* \leq \|\mathbf{\Sigma}\|_* \tag{140}$$

Thus, according to the definition of the nuclear norm, we have:

$$\mathrm{Tr}\left(\Lambda_c\right) \leq \mathrm{Tr}\left(\Lambda\right) \tag{141}$$

Therefore:

$$\lim_{\epsilon \to 0} \frac{\sum_{\boldsymbol{z}_i \in \mathcal{D}_{y=c}^z} \int p_\theta(\boldsymbol{z}|y=c) \log \frac{p_\theta(\boldsymbol{z}|y=c)}{q(\boldsymbol{z},\boldsymbol{z}_i,\epsilon)} \, d\boldsymbol{z}}{\sum_{\boldsymbol{z}_i \in \mathcal{D}_{y=c}^z} \int p_\theta(\boldsymbol{z}) \log \frac{p_\theta(\boldsymbol{z})}{q(\boldsymbol{z},\boldsymbol{z}_i,\epsilon)} \, d\boldsymbol{z}} \tag{142}$$

$$\Rightarrow \lim_{\epsilon \to 0} \frac{\sum_{\boldsymbol{z}_i \in \mathcal{D}_{y=c}^z}\left[\frac{(\boldsymbol{z}_i - \boldsymbol{\mu}_c)^{\top}(\boldsymbol{z}_i - \boldsymbol{\mu}_c)}{\epsilon} - \log\frac{\det(\Lambda_c)}{\epsilon^d} + \frac{\mathrm{Tr}(\Lambda_c)}{\epsilon} - d\right]}{\sum_{\boldsymbol{z}_i \in \mathcal{D}_{y=c}^z}\left[\frac{(\boldsymbol{z}_i - \boldsymbol{\mu})^{\top}(\boldsymbol{z}_i - \boldsymbol{\mu})}{\epsilon} - \log\frac{\det(\Lambda)}{\epsilon^d} + \frac{\mathrm{Tr}(\Lambda)}{\epsilon} - d\right]} \tag{143}$$

Using L'Hospital's rule:

$$\lim_{\epsilon \to 0} \frac{\sum_{\boldsymbol{z}_i \in \mathcal{D}_{y=c}^z}\left[-\frac{(\boldsymbol{z}_i - \boldsymbol{\mu}_c)^{\top}(\boldsymbol{z}_i - \boldsymbol{\mu}_c)}{\epsilon^2} + \frac{d}{\epsilon} - \frac{\mathrm{Tr}(\Lambda_c)}{\epsilon^2}\right]}{\sum_{\boldsymbol{z}_i \in \mathcal{D}_{y=c}^z}\left[-\frac{(\boldsymbol{z}_i - \boldsymbol{\mu})^{\top}(\boldsymbol{z}_i - \boldsymbol{\mu})}{\epsilon^2} + \frac{d}{\epsilon} - \frac{\mathrm{Tr}(\Lambda)}{\epsilon^2}\right]} \tag{144}$$

$$= \frac{\sum_{\boldsymbol{z}_i \in \mathcal{D}_{y=c}^z}(\boldsymbol{z}_i - \boldsymbol{\mu}_c)^{\top}(\boldsymbol{z}_i - \boldsymbol{\mu}_c) + \mathrm{Tr}\left(\Lambda_c\right)}{\sum_{\boldsymbol{z}_i \in \mathcal{D}_{y=c}^z}(\boldsymbol{z}_i - \boldsymbol{\mu})^{\top}(\boldsymbol{z}_i - \boldsymbol{\mu}) + \mathrm{Tr}\left(\Lambda\right)} \tag{145}$$

$$\leq 1 \tag{146}$$

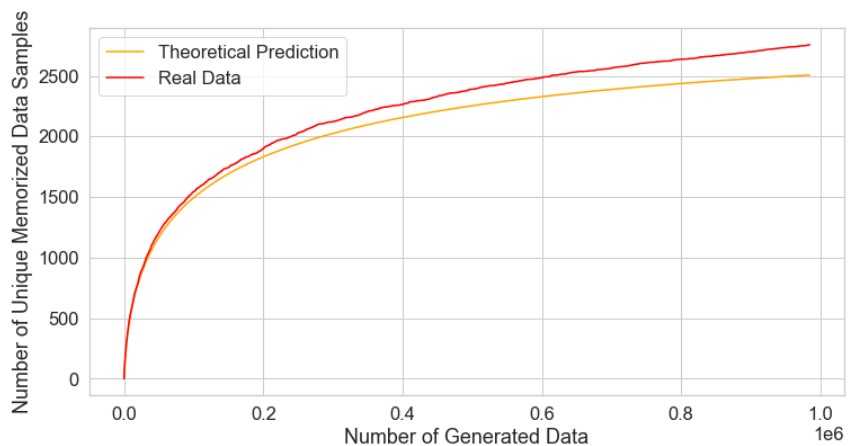

Figure 5: Validation of $N_G$'s significance

## B  EXPLANATION OF THE $N_G$'S IMPORTANCE IN FAIR COMPARISON

By highlighting the significance of $N_G$, UMS allows a fair comparison among extraction methods. We explain this both theoretically and experimentally. Let $N_{\text{umem}}$ denote the number of uniquely memorized images, $M$ the dataset size, and $p_\gamma(i)$ the probability that image $i$ is memorized. To find $\mathbb{E}(N_{\text{umem}})$, we introduce a new variable $I_i$, representing the generation of image $i$ in $N_G$ trials:

$$p(I_i) = 1 - (1 - p_\gamma(i))^{N_G} \tag{147}$$

Using the linearity of expectation, we derive: $\mathbb{E}\left(N_{\text{mem}}\right)$

$$\mathbb{E}\left(N_{\text{umem}}\right) \tag{148}$$
$$= \mathbb{E}\left(I_1\right) + \mathbb{E}\left(I_2\right) + \cdots + \mathbb{E}\left(I_M\right) \tag{149}$$
$$= \sum_{i=1}^{M} 1 - \left(1 - p_\gamma\left(i\right)\right)^{N_G} \tag{150}$$

The importance of $N_G$ lies in its impact on the non-linear expectation of uniquely memorized images. Comparing UMS across different $N_G$ values is flawed because varying $N_G$ leads to different outcomes, underscoring the need for consistent $N_G$ values to ensure fair comparisons. In Carlini et al. (2023), the significance of $N_G$ was overlooked, as they only reported the number of uniquely extracted images. Our theoretical analysis accurately aligns with the behavior observed in the experimental data in Fig 5.

## C  REFINEMENT RESNET BLOCK

The integration of the time module directly after batch normalization within the network architecture is a reasonable design choice rooted in the functionality of batch normalization itself. Batch normalization standardizes the inputs to the network layer, stabilizing the learning process by reducing internal covariate shifts. By positioning the time module immediately after this normalization process, the model can introduce time-dependent adaptations to the already stabilized features. This placement ensures that the temporal adjustments are applied to a normalized feature space, thereby enhancing the model's ability to learn temporal dynamics effectively.

Moreover, the inclusion of the time module at a singular point within the network strikes a balance between model complexity and temporal adaptability. This singular addition avoids the potential redundancy and computational overhead that might arise from multiple time modules. It allows the network to maintain a streamlined architecture while still gaining the necessary capacity to handle time-varying inputs.

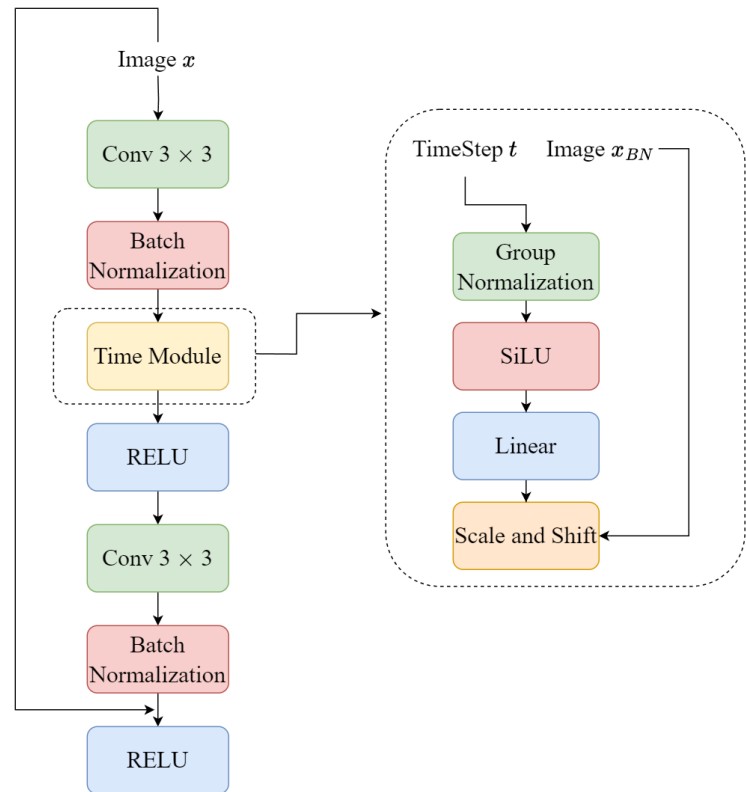

Figure 6: Refinement ResNet block with time-dependent module integration. This block diagram depicts the insertion of a time module within a conventional ResNet block architecture, allowing the network to respond to the data's timesteps. Image $x_{BN}$ is the image processed after the first Batch Normalization Layer.

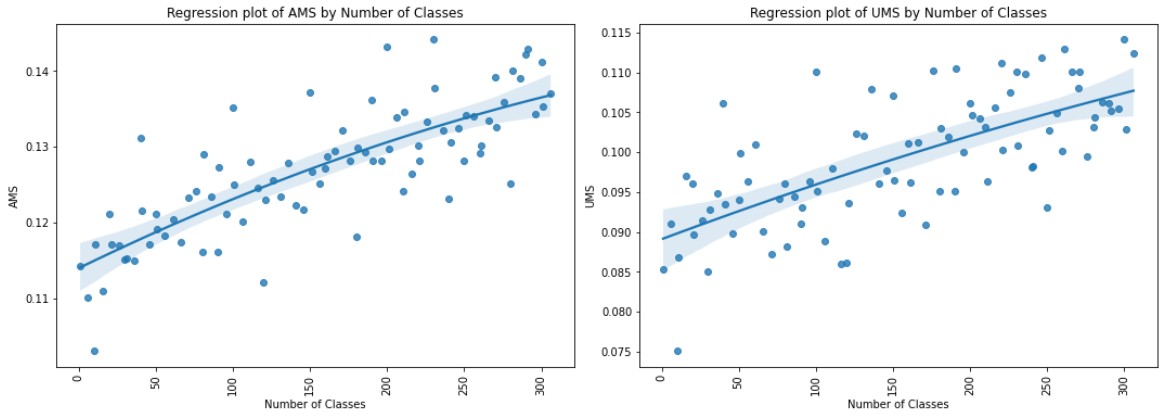

Figure 7: Scatter plots showing the relationship between the number of classes and two performance metrics, AMS (left) and UMS (right). The fitted regression lines demonstrate a positive correlation in both cases.

## D    RESULTS ON CLASSIFIER CHOICE

The analysis of classifier performance across varying numbers of classes reveals interesting patterns for both AMS and UMS. As shown in Figure 7, the scatter plots highlight a positive relationship

between the number of classes and the performance metrics. For AMS, the regression line suggests a stronger relationship ($R^2 = 0.637$), implying that as the number of classes increases, the AMS metric improves with a moderately strong association. In contrast, the relationship between the number of classes and UMS, while still positive, exhibits a slightly weaker connection ($R^2 = 0.483$).

These results suggest that classifier performance, particularly as measured by AMS, benefits more significantly from an increase in the number of classes compared to UMS. The shaded regions in the plots represent the 95% confidence intervals, indicating the range of uncertainty around the fitted regression lines. Overall, the findings imply that the choice of classifier could have a notable impact on AMS , with a less pronounced but still meaningful effect on UMS.

# E  PSEUDOCODE FOR SIDE METHOD

The following Algorithm 1 outlines the detailed steps of the SIDE method for extracting training data from unconditional diffusion models. This algorithm combines the construction of implicit informative labels, the training of a time-dependent classifier, and the conditional generation process to effectively extract valuable training samples.

**Algorithm 1** SIDE Method for Extracting Training Data from Unconditional Diffusion Models

1: **Input:**
  - Unconditional Diffusion Probabilistic Model (DPM) $p_\theta(\boldsymbol{x}) = \mathcal{N}(\boldsymbol{\mu}, \Sigma_{p_\theta})$
  - Pre-trained Classifier $p_\theta(y_I \mid \boldsymbol{x})$
  - Hyperparameter set $\mathcal{S}_\lambda$
  - Number of generated samples per $\lambda$, $N_G$
  - Target label $y = c$

2: **Output:** Extracted training data $\mathcal{D}_{extracted}$

3: **Construct Implicit Informative Labels**

4: **Input:** Unconditional DPM $p_\theta(\boldsymbol{x})$, Classifier $p_\theta(y_I \mid \boldsymbol{x})$

5: Define sampling process with implicit labels:

$$\mathrm{d}\boldsymbol{x} = \left[ f(\boldsymbol{x}, t) - g(t)^2 \left( \nabla_{\boldsymbol{x}} \log p_\theta^t(\boldsymbol{x}) + \lambda \nabla_{\boldsymbol{x}} \log p_\theta^t(y_I \mid \boldsymbol{x}) \right) \right] \mathrm{d}t + g(t)\mathrm{d}\boldsymbol{w} \tag{151}$$

6: **Calibrate Classifier Output**

7: Adjust classifier probabilities using power prior:

$$p_\theta^t(\boldsymbol{x} \mid y_I) \propto p_\theta^{t\lambda}(y_I \mid \boldsymbol{x}) \, p_\theta^t(\boldsymbol{x}) \tag{152}$$

8: **Train Time-Dependent Classifier via TDKD**

9: **Input:** Pre-trained Classifier $p_\theta(y_I \mid \boldsymbol{x})$, Synthetic Dataset $\mathcal{D}_{synthetic}$

10: **Output:** Time-Dependent Classifier $p_\theta^t(y_I \mid \boldsymbol{x}_t)$

11: Initialize Time-Dependent Classifier architecture with time-dependent modules

12: Generate synthetic dataset using DPM:

$$\mathcal{D}_{synthetic} = \{\boldsymbol{x}^{(i)}\}_{i=1}^{N_{synthetic}} \sim p_\theta(\boldsymbol{x}) \tag{153}$$

13: Generate pseudo labels using pre-trained classifier:

$$y_I^{(i)} = p_\theta(y_I \mid \boldsymbol{x}^{(i)}) \tag{154}$$

14: Train Time-Dependent Classifier by minimizing KL divergence:

$$\mathcal{L}_{distil} = D_{KL}\left( p_\theta(y_I \mid \boldsymbol{x}^{(i)}) \,\|\, p_\theta^t(y_I \mid \boldsymbol{x}_t^{(i)}) \right) \tag{155}$$

15: **Overall SIDE Procedure**

16: **Input:** Trained Time-Dependent Classifier $p_\theta^t(y_I \mid \boldsymbol{x}_t)$, Target Label $y = c$, Hyperparameter set $\mathcal{S}_\lambda$, Number of samples $N_G$

17: Initialize empty dataset $\mathcal{D}_{extracted}$

18: **for** each $\lambda \in \mathcal{S}_\lambda$ **do**

19:     **for** $i = 1$ to $N_G$ **do**

20:         Sample $\boldsymbol{x}_0$ conditioned on $y = c$ using Eq. (151)

21:         Compute gradient of cross-entropy loss:

$$\nabla_{\boldsymbol{x}_t} \mathcal{L}_{CE}(c, p_\theta^t(y \mid \boldsymbol{x}_t)) \tag{156}$$

22:         Reverse diffusion process using computed gradient

23:         Generate similarity score for $\boldsymbol{x}_t$

24:         Append $\boldsymbol{x}_t$ to $\mathcal{D}_{extracted}$ with similarity score

25:     **end for**

26: **end for**

27: **Evaluate Attack Performance**

28: Compute evaluation metrics on $\mathcal{D}_{extracted}$

29: **return** $\mathcal{D}_{extracted}$

# F SSCD-RELATED METRICS

In our evaluation of the proposed method, it is essential to include a discussion of the 95th percentile SSCD metric alongside our newly introduced metrics, AMS (Average Matching Similarity) and UMS (Unconditional Matching Similarity). While 95th percentile SSCD metric has its limitations, it still serves as a useful reference point for assessing the relative similarity between extracted images and the original training dataset.

As shown in Table 4, we provide a comprehensive comparison of performance across different datasets.

In addition, we report detailed metrics in the context of our experiments on the CelebA-HQ-FI dataset, which further illustrates the effectiveness of our approach. These results underscore the significance of incorporating SSCD-related scores to complement AMS and UMS in providing a more nuanced understanding of the similarities between generated and training images.

Table 3: Performance Comparison on CelebA-HQ-FI and CelebA-25000 Datasets

| Dataset | Method | Top 0.01% | Top 0.05% | Top 0.1% | Top 0.5% | Top 1.0% | Top 5.0% |
|---|---|---|---|---|---|---|---|
| CelebA-HQ-FI | Uncond | 0.656 | 0.624 | 0.604 | 0.544 | 0.518 | 0.463 |
| | SIDE(ours) | **0.680** | **0.639** | **0.618** | **0.567** | **0.544** | **0.485** |
| CelebA-25000 | Uncond | 0.565 | 0.538 | 0.525 | 0.491 | 0.475 | 0.434 |
| | SIDE(ours) | **0.585** | **0.554** | **0.539** | **0.506** | **0.491** | **0.450** |

Table 4: Generate Training Epoch: 3000 Dataset: CelebA-HQ-FI Generate Nums Per $\lambda$: 50000. The AMS and UMS is measured on Mid Similarity

| $\lambda$ | AMS(%) | UMS(%) | Top 0.1% | Top 0.5% | Top 1.0% | Top 5.0% | Top 10.0% |
|---|---|---|---|---|---|---|---|
| 0 | 0.596 | 0.328 | 0.604 | 0.544 | 0.518 | 0.463 | 0.440 |
| 1 | 0.588 | 0.312 | 0.596 | 0.540 | 0.517 | 0.463 | 0.440 |
| 2 | 0.640 | 0.350 | 0.591 | 0.541 | 0.518 | 0.465 | 0.441 |
| 3 | 0.764 | 0.386 | 0.594 | 0.549 | 0.525 | 0.470 | 0.446 |
| 4 | 0.850 | 0.390 | 0.604 | 0.553 | 0.529 | 0.473 | 0.448 |
| 5 | 0.952 | 0.436 | 0.596 | 0.551 | 0.530 | 0.476 | 0.451 |
| 6 | 1.092 | 0.414 | 0.611 | 0.560 | 0.536 | 0.480 | 0.454 |
| 7 | 1.110 | 0.446 | 0.607 | 0.562 | 0.539 | 0.482 | 0.457 |
| 8 | 1.148 | 0.444 | **0.618** | 0.566 | 0.542 | 0.484 | 0.458 |
| 9 | 1.274 | **0.478** | 0.615 | 0.567 | 0.544 | 0.485 | 0.459 |
| 10 | 1.338 | 0.444 | 0.613 | 0.569 | 0.546 | 0.487 | 0.461 |
| 11 | 1.292 | 0.454 | 0.604 | 0.562 | 0.541 | 0.486 | 0.460 |
| 12 | 1.262 | 0.406 | 0.617 | 0.567 | 0.544 | 0.486 | 0.460 |
| 13 | 1.390 | 0.432 | 0.617 | 0.569 | 0.546 | 0.489 | 0.462 |
| 14 | 1.232 | 0.384 | 0.613 | 0.567 | 0.544 | 0.485 | 0.459 |
| 15 | **1.516** | 0.462 | 0.616 | **0.570** | **0.548** | **0.490** | **0.463** |
| 16 | 1.280 | 0.390 | 0.612 | 0.566 | 0.543 | 0.487 | 0.461 |
| 17 | 1.282 | 0.386 | 0.605 | 0.561 | 0.541 | 0.486 | 0.460 |
| 18 | 1.330 | 0.374 | 0.616 | 0.569 | 0.545 | 0.488 | 0.461 |
| 19 | 1.204 | 0.354 | 0.612 | 0.564 | 0.541 | 0.485 | 0.460 |
| 20 | 1.178 | 0.358 | 0.603 | 0.559 | 0.538 | 0.483 | 0.458 |
| 21 | 1.172 | 0.342 | 0.617 | 0.566 | 0.542 | 0.484 | 0.459 |
| 22 | 1.208 | 0.368 | 0.602 | 0.560 | 0.539 | 0.485 | 0.459 |
| 23 | 1.286 | 0.302 | 0.607 | 0.561 | 0.540 | 0.485 | 0.459 |
| 24 | 1.244 | 0.352 | 0.597 | 0.558 | 0.538 | 0.484 | 0.458 |
| 25 | 1.198 | 0.340 | 0.599 | 0.560 | 0.538 | 0.483 | 0.458 |
| 26 | 1.220 | 0.338 | 0.601 | 0.559 | 0.539 | 0.483 | 0.458 |
| 27 | 1.128 | 0.320 | 0.608 | 0.561 | 0.538 | 0.483 | 0.457 |
| 28 | 1.102 | 0.314 | 0.604 | 0.556 | 0.534 | 0.481 | 0.456 |
| 29 | 1.034 | 0.290 | 0.595 | 0.556 | 0.534 | 0.481 | 0.456 |
| 30 | 1.026 | 0.326 | 0.602 | 0.557 | 0.535 | 0.480 | 0.455 |
| 31 | 1.020 | 0.268 | 0.591 | 0.551 | 0.531 | 0.479 | 0.455 |
| 32 | 1.054 | 0.282 | 0.593 | 0.551 | 0.531 | 0.479 | 0.455 |
| 33 | 1.106 | 0.306 | 0.600 | 0.555 | 0.535 | 0.481 | 0.456 |
| 34 | 1.062 | 0.288 | 0.582 | 0.547 | 0.529 | 0.479 | 0.454 |
| 35 | 0.922 | 0.266 | 0.587 | 0.547 | 0.527 | 0.477 | 0.453 |
| 36 | 0.874 | 0.260 | 0.585 | 0.545 | 0.525 | 0.477 | 0.453 |
| 37 | 0.964 | 0.258 | 0.589 | 0.549 | 0.528 | 0.477 | 0.452 |
| 38 | 0.888 | 0.246 | 0.582 | 0.543 | 0.524 | 0.475 | 0.452 |
| 39 | 0.940 | 0.274 | 0.587 | 0.548 | 0.528 | 0.476 | 0.452 |
| 40 | 0.808 | 0.234 | 0.587 | 0.544 | 0.524 | 0.474 | 0.451 |
| 41 | 0.870 | 0.252 | 0.582 | 0.543 | 0.524 | 0.476 | 0.452 |
| 42 | 0.872 | 0.238 | 0.584 | 0.543 | 0.523 | 0.475 | 0.451 |
| 43 | 0.856 | 0.244 | 0.584 | 0.545 | 0.525 | 0.475 | 0.451 |
| 44 | 0.796 | 0.212 | 0.578 | 0.540 | 0.521 | 0.473 | 0.449 |
| 45 | 0.770 | 0.242 | 0.580 | 0.538 | 0.519 | 0.472 | 0.449 |
| 46 | 0.774 | 0.218 | 0.580 | 0.540 | 0.521 | 0.472 | 0.448 |
| 47 | 0.754 | 0.214 | 0.581 | 0.542 | 0.521 | 0.471 | 0.448 |
| 48 | 0.716 | 0.218 | 0.572 | 0.536 | 0.518 | 0.471 | 0.448 |
| 49 | 0.694 | 0.216 | 0.570 | 0.533 | 0.515 | 0.469 | 0.446 |
| 50 | 0.728 | 0.204 | 0.576 | 0.535 | 0.518 | 0.471 | 0.447 |

