# OpenReview forum: "Towards a Theoretical Understanding of Memorization in Diffusion Models"
_ICLR.cc/2025/Conference — ICLR 2025 Conference Withdrawn Submission_

### Official Review · Reviewer_xP6A · 2024-11-01

**Soundness:** 1
**Presentation:** 2
**Contribution:** 3
**Rating:** 1
**Confidence:** 3

**Summary:**

It has been observed that conditional generative models tend to be more prone to memorization than unconditional models. The authors attempt to show that this will hold generally under some theoretical assumptions. Inspired by this result, they propose a method of extracting memorized samples from generative models by first inducing conditioning on that model. The induced conditioning then causes the model to generate more memorized samples over baseline methods.

**Strengths:**

The idea of inducing conditionality on an unconditioned model to encourage the model to generate memorized images is quite clever.

**Weaknesses:**

I believe the paper is incorrect.

Unless I’m missed some assumption, proposition 1 is incorrect. Looking at the proof, it looks like a mistake occurs in eq78: the left hand side does not equal the covariance matrix of Z|Y since the expectation is taken over all of Z instead of Z|Y. Also a counterexample to the proposition exists: you can increase the variance of a set of points by removing samples at the mean.

I don’t see how Theorem 1 could possibly be correct in the general case. For example, let’s say a line in semantic space is mapped to a circle in sample space. Then if one travels along the straight line in semantic space, they will always recede in semantic space, while in sample space they will recede initially, but eventually return. The proof also appears to be problematic, in particular the change of variable looks incorrect (the Jacobians should definitely not cancel out when you perform a change of variable).

Again, perhaps there was some assumption I missed conditioned on which these theorems are true - if the authors can satisfactorily address this I am of course willing to reevaluate this paper. Otherwise, these errors cast serious doubts on the claims of this paper.

**Questions:**

See weaknesses - if I am mistaken about the technical flaws of this paper then I will be more than happy to reevaluate.

Why are the AMS and UMS metrics not cumulative? It is a little difficult to interpret what low/medium similarity AMS/UMS means, since a sample might be hiding a high similarity match due to the thresholding, which would change one’s conclusion about memorization.

---

### Official Review · Reviewer_YPNZ · 2024-11-01

**Soundness:** 3
**Presentation:** 1
**Contribution:** 1
**Rating:** 3
**Confidence:** 3

**Summary:**

The authors aim to provide a theoretical perspective on memorization, but much of the theory presented feels somewhat elementary (to my understanding, everything is proven with the assumption that $p_\theta$ is a mixture of multivariate normal), with proofs that seem unnecessarily bloated, and the main metric for memorization is entirely inconsequential from section 4 onwards; I will add more detail about this in the weakness section. While I would not recommend the paper for acceptance at this stage, I believe it has promise and could be strengthened significantly by shifting towards a more practical approach with an increased focus on experiments to enhance impact.

In particular, the authors introduce a promising guidance-based method for identifying memorized datapoints (although I do have some questions about its practical validity which I will explain in the questions section).
Unfortunately, the decision to frame the paper primarily as a theoretical contribution has somewhat overshadowed this approach.
Specifically, while the authors define a metric, $\mathcal{M}{point}$, in Definition 1 and aim to explain how increased specificity in the input distribution around a cluster might lead to memorization, this metric is not actually used in practice. The central message of Theorem 2 could be conveyed more succinctly without the excess terminology. Ultimately, the authors do not even rely on $\mathcal{M}{point}$ but revert to using the standard L2 and SSCD similarity metrics, and as a reader, this was quite disappointing, given the time and effort required to work through the "theoretical" sections.

Regarding the experimental setup, I have a few concerns. The authors introduce two metrics, AMS and UMS, based on SSCD and L2 distances, which only outperform the random baseline. There are, however, more advanced methods available. For instance, Wen et al. [A] offer an insightful metric by examining the classifier-free guidance norm, and Ross et al. [B] propose using the score norm along with another geometric measure for detecting memorization. A comparison with at least one of these methods would add valuable context. Most importantly, the metrics that SIDE beats the baselines, AMS and UMS, are proposed **by the authors themseleves**. Although they explain why such metrics have their limitation, it is unclear whether or not the hyperparameters in AMS and UMS have been specifically tuned to get the best performance gap. Adding a comparison, such as using the 95th percentile of SSCD, or qualitative evidence showing previous metrics' limitations would resolve this concern.

Lastly, much of the memorization literature focuses on pre-trained Stable Diffusion models on extensive LAION datasets. It may be worthwhile to consider including experiments on prompts from sources like Webster et al. [C] and Wen et al. [A] to align the analysis with prior work.

### References

[A] Wen, Yuxin, et al. "Detecting, explaining, and mitigating memorization in diffusion models." The Twelfth International Conference on Learning Representations. 2024.

[B] Ross, Brendan Leigh, et al. "A geometric framework for understanding memorization in generative models." arXiv preprint arXiv:2411.00113 (2024)

[C] Webster, Ryan. "A reproducible extraction of training images from diffusion models." arXiv preprint arXiv:2305.08694 (2023).

**Strengths:**

1. Using guidance on an unconditional DPM to determine whether or not an image is memorized is a smart approach!
2. I do appreciate the attention to detail in the proofs; even though they are sometimes overly long.

**Weaknesses:**

Theoretical side:

1. The authors appear to assume $p_\theta$ is a multivariate normal throughout the proofs, which seems extremely inappropriate and is obscured by the theorem presentation. Please clarify if this is incorrect.

2. Eq. (1) is basically the sum of KL divergences between Gaussians centred at training points and the learned distribution. Let's assume that the training datapoints are well apart and the learned distribution is a mixture of Gaussians centred at each of the training datapoints (perfect memorization). Now let us take the limit $\epsilon \to 0$, while at around the memorized point, the term $p_\theta / q(x, x_i, \epsilon)$ converges to $1/|\mathcal{D}|$ at all other points it will blow up to infinity. Therefore, as $\epsilon \to 0$ the metric blows up to infinity and it cannot detect an obvious case of memorization. The metric, in and of itself, is useless (which is probably why the authors do not use it in practice). However, it can aid in proving Theorem 2, which is trying to show a much easier phenomenon: increased specificity in the input distribution around a cluster might lead to memorization.

3. The authors use the term "suffix training" without defining what it means. To my understanding, it follows from (6), which is basically maximum likelihood across all the conditions -- which is how conditional DPMs are trained --- but there is no explanation whatsoever, and there is no reason to believe that unconditional DPMs with "implicit" labels are trained this way.

4. What is the definition of the norm used in Eq. (8)? Upon reading it seems it is the matrix norm induced by the L2 norm, but it is not defined in the definition.

Practical Approach:

1. The metrics seem invented (AMS/UMS) and tuned with strange hyperparameters $\alpha$ and $\beta$. At least one additional metric which is widely accepted is required to determine whether SIDE is a valid approach.
2. The only baseline that is compared against is "random", at least the baselines from Wen et al. [A] or Ross et al. [B] should be added to the analysis to make SIDE convincing.

**Questions:**

Indeed, one may get a false positive from the SIDE method if one has access to an extremely good classifier that always guides the generation towards the input image. The authors have a counter-argument for this in "Memorized Images are not from the classifiers", but I cannot follow the rationale, could you please explain more?

---

### Official Review · Reviewer_YPKU · 2024-11-01

**Soundness:** 3
**Presentation:** 4
**Contribution:** 3
**Rating:** 6
**Confidence:** 3

**Summary:**

The study aims to understand and measure memorization in diffusion probabilistic models (DPMs), a critical area given the risks of data leakage and privacy concerns associated with these models. The authors propose a theoretical framework to explain why conditional DPMs tend to memorize more training data than unconditional DPMs, introducing the concept of "informative labels" which enhance memorization by enabling tighter clustering in the latent space. This framework supports the creation of a new method for training data extraction called Surrogate Conditional Data Extraction (SIDE), which uses a time-dependent classifier to simulate conditions that facilitate data extraction even from unconditional DPMs.

**Strengths:**

The paper has the following strengths:

1. Novelty of Theoretical Contributions: The paper offers a substantial theoretical advancement in understanding memorization within diffusion probabilistic models (DPMs), both conditional and unconditional. The authors introduce a novel metric to measure point-wise memorization and extend their analysis with a theoretical framework that differentiates between memorization patterns in these models.

2. Empirical Validation: The experimental section robustly supports the theoretical claims. By implementing a novel method named SIDE (Surrogate condItional Data Extraction), the authors effectively demonstrate its utility in extracting data from models in challenging scenarios. The results are presented clearly and are compelling, especially the significant improvements in performance metrics over previous methods.

**Weaknesses:**

The paper has the following weaknesses:

1. Typos and Formatting Errors: The paper contains several minor typographical errors, such as incorrect numbering and missing spaces (e.g., "line 045" and "line 367"). These errors, while minor, could detract from the paper's professional quality and should be corrected to maintain the paper's credibility.

2. Clarification Needed on Downstream Impacts: The connection between the proposed method's impact on memorization and its benefits for downstream task performance (e.g., improved FID scores in image generation) is not well-established. Clarifying whether SIDE enhances performance in practical applications would strengthen the paper's impact and relevance.

**Questions:**

1. Clarification on Random Labels: The paper suggests that conditioning on random labels increases memorization, but it's counterintuitive as it might introduce incorrect data associations. Could the authors provide a more intuitive explanation or direct evidence to support this claim beyond theoretical analysis?

2.  Impact on Downstream Tasks: Does the SIDE method improve downstream task performance, such as FID scores in generative tasks? It would be beneficial to see empirical evidence or a discussion on how memorization impacts practical outputs in real-world applications.

---

### Official Review · Reviewer_uuRM · 2024-11-04

**Soundness:** 2
**Presentation:** 3
**Contribution:** 2
**Rating:** 5
**Confidence:** 4

**Summary:**

This paper investigates the memorization of training data in diffusion probabilistic models (DPMs), which are widely used in generative AI. While previous studies focused on conditional DPMs and found them more prone to memorization, extracting data from unconditional models has been challenging. The authors provide a theoretical analysis showing that data extraction from unconditional DPMs is feasible by constructing a surrogate condition. They introduce a method called Surrogate Conditional Data Extraction (SIDE), which leverages a time-dependent classifier trained on generated data to extract training data from unconditional models.

**Strengths:**

1. The paper is logically structured, with each section clearly written and the language fluent and easy to understand.

2. The theoretical proofs are presented quite clearly. While I did not thoroughly verify every proof, I found the proofs in the appendix easy to follow and believe they are correct.

**Weaknesses:**

1. The theoretical part of the paper does not consider exact diffusion models but instead focuses on an encoder-decoder pair trained via maximum likelihood estimation. I believe there is a gap here that needs to be addressed. The authors' setting significantly simplifies score-based models. While I understand this simplification aids in proving theoretical results, discussing this gap is necessary.

2. There is a disconnect between the theoretical analysis and the proposed method. The theory mainly focuses on how informative labels (explicit or implicit) can lead to stronger memorization of training data (as discussed in Proposition 1 and Theorem 2). However, the SIDE method primarily aims to extract information from unconditional DPMs. This paper implicitly assumes that the training process of unconditional DPMs extracts classification information from the training data, forming implicit informative labels and lead to memorization. While this may be well-known from the authors' perspective (e.g., references [1] and [2]), I believe this logical premise needs to be explicitly discussed.

3. The experimental section lacks direct validation of the theoretical findings. Attacking unconditional DPMs by artificially learning class information of the training data does not convincingly demonstrate that unconditional DPMs acquire memorization through implicitly learning data labels. Since additional classifier information is introduced, such attacks may result from data leakage rather than solely from the memorization of DPMs. Therefore, additional experiments are necessary. I suggest that the authors should at least verify the impact of different labels on conditional DPMs using a small-scale dataset like CIFAR-10, exploring scenarios with normal labels, clustered labels (as in reference [3]), etc.

4. The setting of the proposed TDKD method is somewhat peculiar. The authors implicitly assume that the training dataset of the DPMs under attack is inaccessible, yet they assume that a pre-trained time-independent classifier on this training dataset is available. This seems like a deliberately crafted special scenario. Why is the time-independent classifier on this dataset available while the time-dependent classifier is not? In the experimental section, the original training data is fully accessible. I do not understand why the authors start from a time-independent classifier and use TDKD for distillation instead of directly training a time-dependent classifier. This raises doubts about the effectiveness of the SIDE attack. In diffusion models trained on non-public data, how can we obtain a time-independent classifier?

5. In the introduction, the authors state that one of their contributions is: "We introduce a novel metric to measure the degree of point-wise memorization in DPMs." However, I did not see the specific measurement methods for Pointwise Memorization and Semantic Memorization in the experimental section. Instead, the authors use Average Memorization Score and Unique Memorization Score. What prevents them from utilizing the metrics proposed in the theoretical section in practical experiments? This is crucial for evaluating the contributions of the theoretical part based on these metrics.

**References:**

1. **Representation Alignment for Generation: Training Diffusion Transformers Is Easier Than You Think**
2. **Your Diffusion Model is Secretly a Zero-Shot Classifier**
3. **Hierarchical Clustering for Conditional Diffusion in Image Generation**

**Questions:**

1. I find the authors' discussion on mitigating memorization in conditional DPMs through label perturbation intriguing. Would this affect the model's training performance? Is there a trade-off involved? However, the authors have not conducted further experiments or research on this aspect.

---

### Note · Authors · 2024-11-14

I have read and agree with the venue's withdrawal policy on behalf of myself and my co-authors.